

# TRANSFER OF RADIOCAESIUM FROM CONTAMINATED BOTTOM SEDIMENTS TO MARINE ORGANISMS THROUGH BENTHIC FOOD CHAIN IN POST-FUKUSHIMA AND POST-CHERNOBYL PERIODS

Roman Bezhenar[1], Kyung Tae Jung[2], Vladimir Maderich[1], Stefan Willemsen[3], Govert de With[3], and Fangli Qiao[4]

[1]Institute of Mathematical Machine and System Problems, Glushkov av., 42, Kiev 03187, Ukraine
[2]Korea Institute of Ocean Science and Technology, 787, Haean-ro, Ansan 426-744 Republic of Korea
[3]NRG, Utrechtseweg 310, 6800 ES Arnhem, the Netherlands
[4]First Institute of Oceanography, 6 Xianxialing Road Qingdao 266061 China

*Correspondence to:* Vladimir Maderich (vladmad@gmail.com)

**Abstract.** After the earthquake and tsunami on 11 March, 2011 damaged the Fukushima Dai-ichi Nuclear Power Plant (FDNPP), an accidental release of a large amount of radioactive isotopes into both the air and the ocean occurred. Measurements provided by the Japanese agencies over the past four years show that elevated concentrations of $^{137}$Cs still remain in sediments, benthic organisms

and demersal fishes in the coastal zone around the FDNPP. These observations indicate that there are $^{137}$Cs transfer pathways from bottom sediments to the marine organisms. To describe the transfer quantitatively, the dynamic food chain model BURN has been extended to include benthic marine organisms. The extended model takes into account both pelagic and benthic marine organisms grouped into several classes based on their trophic level and type of species: phytoplankton, zooplankton, and

fishes (two types: piscivorous and non-piscivorous) for the pelagic food chain; deposit feeding invertebrates, demersal fishes feeding by benthic invertebrates and bottom omnivorous predators for the benthic food chain; crustaceans, molluscs and coastal predators feeding on both pelagic and benthic organisms. Bottom invertebrates ingest organic parts of bottom sediments with adsorbed radionuclides which then migrate up through the food chain. All organisms take radionuclides directly

from water as well as food. The model was implemented into the compartment model POSEIDON-R and applied to the Northwestern Pacific for the period of 1945-2010 and then for the period of 2011-2020 to assess the radiological consequences of releases of $^{137}$Cs due to FDNPP accident. The model simulations for activity concentrations of $^{137}$Cs in both pelagic and benthic organisms in the coastal area around the FDNPP agree well with measurements for the period of 2011-2015. The de-

crease constant in the fitted exponential function of simulated concentration for the deposit ingesting



invertebrates (0.45 y$^{-1}$) is close to the decrease constant for the sediment observations (0.44 y$^{-1}$), indicating that the gradual decrease of activity in the demersal fish (decrease constant is 0.46 y$^{-1}$) was caused by the transfer of activity from organic matter deposited in bottom sediment through the deposit feeding invertebrates. The estimated from model transfer coefficient from bulk sediment to

demersal fish in the model for 2012-2020 (0.13) is larger than that to the deposit feeding invertebrates (0.07) due to the biomagnification effect. In addition, the transfer of $^{137}$Cs through food webs for the period of 1945-2020 has been modelled for the Baltic Sea that was essentially contaminated due to global fallout and the Chernobyl accident. The model simulation results obtained with generic parameters are also in good agreement with available measurements in the Baltic Sea. Due to weak

water exchange with the North Sea of the semi-enclosed Baltic Sea the chain of water-sediments-biota slowly evolves into a quasi-equilibrium state unlike the processes off the open Pacific Ocean coast where the FDNPP is located. Obtained results demonstrate the importance of the benthic food chain in the long-term transfer of $^{137}$Cs from contaminated bottom sediments to marine organisms and the potential of a generic model for the use in different regions of the World Ocean.

## 35    1    Introduction

A catastrophic earthquake and tsunami that occurred on 11 March, 2011 severely damaged the Fukushima Dai-ichi Nuclear Power Plant (FDNPP). The loss of power and the subsequent overheating, meltdowns, and hydrogen explosions at the FDNPP site resulted in the uncontrolled release of radioactivity into the air and ocean (Povinec et al., 2013). The atmospheric fallout over the land

and the ocean peaked in mid-March whereas the direct release to the ocean from FDNPP peaked in the beginning of April. Approximately 80% of the radioactivity released due to the accident in March-April 2011 was either directly discharged into the ocean or deposited onto the ocean surface from the atmosphere. The concentration of $^{137}$Cs in the ocean reached a maximum in mid-April of 2011 and has thereafter declined (by a factor of $10^5$ ), except for the area around the FDNPP, where

continuous leaks of contaminated water have been reported (Kanda, 2013). However, the concentration of $^{137}$Cs in the bottom sediment that was contaminated by water with high concentrations in April-May 2011 remains quite high and is showing signs of very slow decrease with time (Otosaka and Kobayashi, 2013; Kusakabe et al. 2013; Ambe et al 2014; Black and Buesseler, 2014). The concentration of organically bound $^{137}$Cs in coastal areas is several times higher than that of the bulk

sediment (Otosaka and Kobayashi, 2013; Ono et al., 2015) due to $^{137}$Cs adsorption on organic matter. It is worth noting that organic content in the shelf of Fukushima and Ibaraki Prefectures varies in the range of 0.1-25% (Otosaka and Kobayashi, 2013; Ambe et al., 2014; Ono et al., 2015). The preferential adsorption of $^{137}$Cs on organic matter can be explained by the partial coverage of fine mineral sediment by organic substances and subsequent blocking of sorption (Kim et al., 2006; Ono

et al., 2015). Comparison of the concentration of $^{137}$Cs in the sediment and benthic invertebrates



(Sohtome et al., 2014) and in the demersal fishes (Buesseler et al., 2012; Wada et al., 2013; Tateda et al., 2013) suggests that the continual ingestion of organic matter from sediments can be an important contamination pathway for all components of the benthic food web. However, in most of the benthic food web models applied to the FDNPP accident, the deposit ingestion is not included as a transfer

process in the food-chain (Tateda et al., 2013; Keum et al., 2015; Vives i Batlle 2015; Tateda et al., 2015a,b; Vives i Batlle et al., 2015a,b).

Several models were used to perform long term assessments of the radiological impact in the marine environment due to the FDNPP accident (Nakano and Povinec, 2012; Maderich et al., 2014a,b). In particular, the compartment model POSEIDON-R (Maderich et al., 2014a,b) correctly predicted

the concentration of $^{137}$Cs and $^{90}$Sr in water and sediments in the coastal box (30x15 km) around the FDNPP for 2011-2013. In that study the flux of radionuclides due to the underground leakage of contaminated waters from FDNPP (Kanda, 2013) was taken into account. However, the version of the dynamic food-chain model BURN (Biological Uptake model of Radionuclides) used in the POSEIDON-R model (Heling et al., 2002; Lepicard et al., 2004, Maderich et al., 2014a,b) did not

take into account the benthic food web. Nevertheless the results of simulations still agreed well with observations for the first months and years when transfer from water dominated (Maderich et al., 2014a,b). Measurements following the Fukushima Dai-ichi accident suggest that transfer of radioactivity from bottom deposits through the benthic food web over a prolonged period of time can be an increasingly important factor in the radiological assessment of released radioactivity.

Another relevant case is the significant contamination of the Baltic Sea in 1986 by the deposition of activity originating from the Chernobyl accident. Unlike the coastal sea region near FDNPP, the Baltic Sea is a semi-enclosed relatively shallow sea filled by brackish waters and connected with the ocean by the narrow and shallow Danish Straits (Leppäranta and Myrberg, 2009). Within HELCOM (Helsinki Convention on the Protection of the Marine Environment of the Baltic Sea

Area, www.helcom.fi) the group MORS (Monitoring of Radioactive Substances) established an internationally agreed monitoring network in 1986 and collected all the data in a common data base (MORS, 2015). Therefore, this event also represents a good test case to validate models (Periañez et al., 2015).

In this study, an extended food web model is presented that considers both pelagic and benthic

foodchains. This dynamic model was implemented into the compartment model POSEIDON-R and applied to the northwestern Pacific for the period of 1945-2020 to assess the radiological consequences from $^{137}$Cs released as a result of global fallout and the Fukushima Dai-ichi accident. The model was also applied to the Baltic Sea for the period 1945-2020 to show the versatile applicability of this model. The paper is organized as follows. Descriptions of the compartment model and of the

extended dynamic food web model are given in Section 2. Section 3 presents the model application for the Fukushima Dai-ichi accident. The results of the model application to the Baltic Sea are given in Section 4. Section 5 summarizes the findings.



## 2 Model description

To describe transfer pathways of $^{137}$Cs from bottom sediments to marine organisms the dynamic
model BURN was extended. The model is based on the approach devised by Heling et al. (2002)
in which the marine organisms are grouped into a few classes based on trophic level and types of
species. The radionuclides are grouped in several classes in terms of tissues in which a specific ra-
dionuclide accumulates preferentially. These simplifications allow for a limited number of standard
input parameters. The scheme of transfer of radionuclides through the marine food web is shown in
Fig. 1. The different food-chains exist in both pelagic and benthic zones. Pelagic organisms are di-
vided into primary producer (phytoplankton) and consumers: zooplankton, forage (non-piscivorous)
fish and piscivorous fish. This food web has been implemented in the compartmental POSEIDON-R
model (Lepicard et al., 2004; Maderich et al., 2014 a,b).

The benthic food web includes three primary pathways for radionuclides: (i) through water con-
tamination in a manner similar to the BURN model, (ii) through the vertical detritus flux and zoo-
plankton faeces (Fowler et al., 1987), and (iii) through contaminated bottom sediments. The radionu-
clides adsorbed on the organic matter in the sediments is bioavailable for benthic organisms but the
mineral component of sediments is not (Ueda et al., 1977; Ueda et al.,1978) although Koyanagi et
al. (1978) found a rapid and more intensive transfer of several sediment adsorbed radionuclides to
particular organs of the demersal fishes. It is assumed that radioactivity is transferred from organic
bottom deposits to deposit feeding invertebrates, then to benthic invertebrate feeding demersal fishes,
and on to omnivorous bottom predators (Fig. 1). The components of this system also include crus-
taceans (e.g detritus-feeders),molluscs (filter-feeders), and coastal predators feeding in the whole
water column in shallow waters.

In the extended model utilised in this study, the concentration of radioactivity in phytoplankton
$C_1$ is calculated using the Biological Concentration Factor (BCF) approach due to the rapid uptake
from water and the short retention time of radioactivity,namely,

$$C_1 = CF_{ph}C_w, \tag{1}$$

where $C_w$ is concentration of radioactivity in water and $CF_{ph}$ the BCF for phytoplankton. For the
macroalgae, a dynamic model is used to describe radionuclide concentrations due to the longer
retention times

$$\frac{dC_5}{dt} = (CF_{ma}C_w - C_5)\ln 2 T_{0.5,5}^{-1}, \tag{2}$$

where $C_5$ is the concentration of radioactivity in the macro-algae, $CF_{ma}$ is corresponding BCF,
$T_{0.5,5}$ is the biological half-life of the radionuclide in the plant and $t$ is the time. The concentration
of a given radionuclide in the zooplankton ($i$=2), invertebrates ($i$=6,7,8) and fish ($i$=3,4,9,10,11; see
Table 1 for a description of the different fish groups in the model) is described by the following





differential equation:

$$\frac{dC_i}{dt} = a_i K_{f,i} C_{f,i} + b_i K_{w,i} C_w - \ln 2 \, T_{0.5,i}^{-1} C_i, \tag{3}$$

where $C_i$ and $C_{f,i}$ are the concentrations of radioactivity in the marine organism and food, respec-
tively, $a_i$ is the food extraction coefficient, $b_i$ is the water extraction coefficient, $K_{f,i}$ is the food
uptake rate, $K_{w,i}$ is the water uptake rate and $T_{0.5,i}$ is the biological half-life of the radionuclide in
the organism.

The activity concentration in the food of a predator $C_{f,i}$ is expressed by the following equation,
summing up for the total of $n$ prey types,

$$C_{f,i} = \sum_{j=0}^{n} C_{prey,j} P_{i,j} \frac{drw_{pred,i}}{drw_{prey,j}}, \tag{4}$$

where $C_{prey,j}$ is the activity concentration in prey of type $j$, $P_{i,j}$ is preference for prey of type $j$,
$drw_{pred,i}$ is the dry weight fraction of predator of type of $i$, and $drw_{prey,j}$ is the dry weight fraction
of prey of type $j$. The index "0" corresponds to the bottom deposit. It is assumed that (i) radioac-
tivity concentration in organic and mineral fractions of bottom deposit are in mutual equilibrium
and (ii) that only organic matter in the bottom deposit is bioavailable. Therefore, the concentration
of assimilated radioactivity from the organic fraction of bottom sediment can be related with the
radioactivity concentration of the upper layer of bulk sediment as $C_{prey,0} = \phi_{org} \cdot C_s$. Here $\phi_{org}$ is
an empirical parameter $\phi_{org} = (1-p) f_{org} C_{org} C_s^{-1}$ where $p$ is porosity, $f_{org}$ is the organic matter
fraction, $C_{org} C_s^{-1}$ is the ratio of concentration $C_{org}$ (Bq kg$^{-1}$-dry) in the organic matter to in the
bulk sediment concentration $C_s$ (Bq kg$^{-1}$-dry). The value of $\phi_{org}$ is in the range 0.1-0.01 (Ono et
al. 2015).

Values of the model parameters are given in Table 1. The parameters for pelagic and benthic food
webs were compiled from published data (Baptist and Price, 1962; Cammen, 1980; De Vries and
De Vries, 1988; Coughtrey and Thorne, 1983; Tateda, 1994,1997; Vives i Batlle et al., 2007; Tateda
et al., 2013; Iwata et al., 2013; Sohtome et al., 2014). The biological half-life data for fish (Baptist
and Price, 1962; Coughtrey and Thorne, 1983; Tateda, 1994,1997; Zhang et al., 2001; Matsumoto
et al., 2015) show variations in a large range (35-270 days) due to the differences between species
and due to the differences in the experiment methodology (Matsumoto et al., 2015). In this generic
model, values of $T_{0.5,i}$ were divided into two groups: $T_{0.5,i} = 75$d for non-piscivorous fishes and
those demersal fishes feeding on invertebrates ($i = 3, 9$) and $T_{0.5} = 150$d for predatory fishes ($i = 4, 10, 11$). This is based on the assumptions that (a) larger fishes have longer $T_{0.5,i}$ due to the slower
metabolic rate (Matsumoto et al., 2015), and (b) predatory fishes are generally larger than prey
fishes. The results of sensitivity study for $T_{0.5,i}$ are given in next section to assess robustness of this
simplification. Additional restriction on the values of the model parameters is the condition that at
equilibrium state BCF of the components of the food chain should be relevant to the values from
IAEA(2004). The values of prey preference are given in Table 2. They are compiled from data on





food habits of organisms (Fujita et al., 1995; Kasamatsu and Ishikawa, 1997; Iwata et al., 2013; Sohtome et al., 2014).

It is well known that the uptake of caesium decreases with increasing salinity due to the increase

of competing ions from potassium with higher concentration. This was taken into account when introducing the salinity-dependent correction factor $F_K$ for phytoplankton and macro-algae since caesium enters the foodweb mainly via the lowest trophic level whereas the uptake from water contributes in a relatively minor way (Heling and Bezhenar, 2009). Based on laboratory experiments with marine plants for caesium, the correction factor was verified against field measurements in the

Dnieper-Boog Estuary (Heling and Bezhenar, 2011). It is expressed as

$$F_K = \frac{0.05}{\exp(0.73\ln(K^+/39.1) - 1.22 \cdot 10^3 \Theta^{-1})}, \tag{5}$$

where $K^+$ is the potassium concentration (mg L$^{-1}$) and $\Theta$ is temperature ($^o$K). For water with a $K^+$ concentration of above 1.5 mg L$^{-1}$, the potassium concentration could be linked to the salinity using the following linear relationship (Heling and Bezhenar, 2009):

$$K^+ = 11.6S - 4.28, \tag{6}$$

where $S$ is the salinity in g L$^{-1}$. Then the BCF for phytoplankton and macro-algae can be expressed by:

$$CF_{ph} = F_K CF_{ph}^*, \qquad CF_{ma} = F_K CF_{ma}^*, \tag{7}$$

where $CF_{ph}^*$=20 Lkg$^{-1}$ and $CF_{ma}^*$=50 Lkg$^{-1}$ are standard BCFs for marine environments (IAEA,

180 2004).

According to a review of radiological data (Coughtrey and Thorne, 1983; Yankovich et al., 2010), every radionuclide in fish accumulates mostly in a specific (target) tissue. It is assumed that the target tissue (bone, flesh, organs and stomach) controls the overall elimination rate of the nuclide ($T_{0.5,i}$) in the organism. The radioactivity in the food for the predator is therefore the activity concentration

in the target tissue diluted by the remaining body mass of the prey fish, calculated by multiplying the predicted level in the target tissue by its weight fraction. For radiocaesium the target tissue is flesh. To calculate the concentration in the edible part of fish from the calculated levels in the target tissues, a target tissue modifier (TTM) is introduced. This is based on tissue distribution information (Coughtrey and Thorne, 1983; Yankovich et al., 2010). Values of the described parameters for the

fish in a dynamic food chain model are given in Table 3.

The dynamic food-chain model is part of the POSEIDON-R (Lepicard et al., 2004; Maderich et al., 2014a,b) compartment model where the marine environment is modelled as a system of compartments representing the water column, bottom sediment and biota. The compartments describing the water column containing suspended matter are subdivided into a number of vertical layers. The

model assumes partition of the radionuclides between the dissolved and particulate fractions in the



water column, described by a distribution coefficient. The radionuclide concentration for each compartment is governed by a set of differential equations including the temporal variations of concentration, the exchange with adjacent compartments and with the suspended and bottom sediments, radioactive sources and decay. The exchange between the water column boxes is described by fluxes

of radionuclides due to advection, sediment settling and turbulent diffusion processes. The activity loss on suspended sediments occurs through settling in underlying compartments and, finally, to the bottom. A three-layer model describes the transfer of radionuclides in the bottom sediment. The transfer of radioactivity from the upper sediment layer to the water column is described by diffusion in the interstitial water and by bioturbation. Radioactivity in the upper sediment layer migrates

downwards by diffusion and by burial at a rate assumed to be the same at which particles settle from the overlying water. The upwards transfer of radioactivity from the middle sediment layer to the top sediment layer occurs only by diffusion. Burial causes an effective loss of radioactivity from the middle to the deep sediment layer, from which no upward migration occurs. The model for the pelagic food web component was implemented for the upper layer of the compartment, whereas the

benthic component was included in the shallow one-layer compartments.

The POSEIDON-R model can handle different types of radioactive releases: atmospheric fallout, runoff from land deposited radionuclide by river systems, point sources associated with routine releases from nuclear facilities located either directly on the coast or inland at river systems, and point sources associated with accidental releases (Lepicard et al., 2004). For coastal discharges occur-

ring in the large ('regional') boxes, 'coastal' release boxes are nested into the regional box system. Advection and diffusion of zooplankton are not taken into account due to the short biological half-life (five days), except in the coastal box, where diffusion was considered. It was assumed that crustaceans, molluscs, and fish are not transported by ocean flows. When calculating the radionuclide concentration in fish in small coastal boxes, random fish migration is taken into account as in

Maderich et al. (2014a,b). For this purpose, the right hand side of equation (3) for radionuclide concentration in fish, both in the inner ($C_{in,i}$) and outer ($C_{out,i}$) compartments, is extended by the term $-(C_{in,i} - C_{out,i})/T_{migr,i}$ for the coastal compartment and by the term $(C_{in,i} - C_{out,i})/(\delta T_{migr,i})$ for the outer compartment. Here $T_{migr,i}$ is the characteristic time of fish migration from a coastal compartment, depending on compartment scale and fish species, and $\delta$ is the ratio between the vol-

umes of the outer and the coastal compartments.

## 3   Application to the Fukushima Dai-ichi accident

### 3.1   Model setup

The model was customized for the northwestern Pacific Ocean, the East China and Yellow Seas and the East/Japan Sea. A total of 176 boxes cover this entire region (Fig. S1). The compartments around

the FDNPP are shown in Fig. 2. The coastal box (15x30 km), which is placed around FDNPP, is



located inside box 90. It was chosen to cover observation data within a circular-shaped surface area with a radius 15 km centred around the FDNPP. The advective and diffusive water fluxes between compartments were based on a ten-year average over the period 2000-2009 using the Regional Ocean Modeling System (ROMS). Details of the customization are given by Maderich et al. (2014a,b). In the simulations the parameters $\phi_{org}$=0.01 and $T_{migr,i}$=0.7 y for $i$=3,4,9,10,11 were used.

The simulation of dispersion and fate of $^{137}$Cs was carried for the period 1945-2010 to provide background concentrations of radiocaesium for the radiological assessment of the FDNPP accident for the period 2011-2020 and to verify the model with available data. The main source of $^{137}$Cs in the northwestern Pacific in the period 1945-2010 was from fallout due to atmospheric nuclear weapon tests. The fallout includes a global component, caused by the transport of radioactivity due to the general atmospheric circulation and subsequent deposition on the surface of the sea and a regional component, caused by fallout from weapon tests carried out in the Marshall Islands, resulting in the contamination of the surface layer of the ocean. The annual deposition of $^{137}$Cs on the ocean for the period 1945-2005, compiled from Nakano (2006) and Hirose et al. (2008), is shown in Fig. S2a. The concentrations of $^{137}$Cs at the eastern and southern boundaries (Fig. S2b) of the computational domain (Fig. S1) were estimated by using both observations from the MARIS (Marine Information System) database (MARIS, 2015), and observations from Kang et al. (1997) and Nakano and Povinec (2003). These values represent both the effect from global deposition of $^{137}$Cs on the northeastern Pacific and the regional effect of weapon tests carried out in the Marshall Islands. For the prediction of the concentration of $^{137}$Cs for the period 2005-2020, five-years-averaged deposition and the boundary concentrations during the period of 2000-2004 were extrapolated and corrected for radioactive decay. The simulation for the period 1945-2010 was continued for the period of 2011-2020 with a source term estimated from the Fukushima accident. It was assumed that the release of activity directly to the ocean took place over the period 1-10 April 2011. Amounts of 5 PBq of $^{134}$Cs, and 4 PBq of $^{137}$Cs were transferred directly into the coastal box. These quantities are in accordance with widely accepted source terms for the Fukushima accident simulations (see Povinec et al., 2013). The atmospheric deposition data was obtained from simulations with the MATCH model (Robertson et al., 1999) where the dispersion of $^{137}$Cs for the period 12 March-5 April was computed (Maderich et al., 2014a). The ECMWF meteorological data with a source term reported by Stohl et al. (2012) was used in the simulation. The amount of deposited $^{137}$Cs in the computational domain was 8.5 PBq. The deposition of $^{134}$Cs was estimated at 10.2 PBq using an activity ratio $^{134}$Cs/$^{137}$Cs=1.2 (NISA, 2011). The atmospheric deposition was distributed between compartments as shown in Fig.2. The continuous leakage into the coastal box from the middle of 2011 with a release rate of 3.6 TBq y$^{-1}$ (Kanda, 2013) was taken into account.



 **3.2 Results**

The results from the modelling of the $^{137}$Cs concentration in the water and in the upper layer of sediments of the coastal box are shown in Fig. 3. Model results for the water demonstrate good agreement both with yearly averaged observations by MEXT (the Japanese Ministry of Education, Culture, Sports, Science and Technology) for the period 1950-2010 (MEXT, 2010) and with obser-

vation by TEPCO (Tokyo Electric Power Company) for the period of 2011-2014 (TEPCO, 2014). Comparison of Fig. 3a with Fig. 9a from Maderich et al. (2014a) confirms that the model correctly simulated the almost constant concentration of $^{137}$Cs in the water in the FDNPP vicinity due to the continued leak of radioactivity from FDNPP (Kanda, 2013). The geometric mean of the simulated-to-observed values is 1.0 for the period 1984-2014 when data were available, with a geometric

standard deviation of 1.9 for a total number of observations $N$=48. The model also predicts well the concentration of $^{137}$Cs in the bottom sediment before the accident and the sudden increase in concentration by more than five orders magnitude as a result of the accident. However, as seen in Fig. 3b, the observed concentration from 2013 decreases faster than the model prediction without including the correction of vertical transfer. The details of this correction are described below. The

estimated decrease constant of the fitted exponential function of the observed sediment concentration for 2012-2014 is $\lambda_s$ = 0.44 y$^{-1}$. The observed concentrations of $^{137}$Cs in the bottom sediment of the coastal areas (B,C,D) with a depths less than 50 m in the Fukushima Prefecture (Sohtome et al., 2014) show a similar decrease. The decrease constant for area B located north of FDNPP is 0.44 y$^{-1}$ whereas for the smaller areas C and D located south of the FDNPP it is 0.63 y$^{-1}$ and 0.7 y$^{-1}$,

respectively. For the deeper offshore area F adjacent to the areas C and D the value of the decrease constant is much less (0.24 y$^{-1}$). Several possible mechanisms could be responsible for the observed time-spatial redistribution of radioactivity in the surface layer of sediment. According to Ambe et al. (2014) the vertical transfer of $^{137}$Cs by resuspension and redeposition by the ocean currents and waves, desorption to the pore water and bioturbation can result in a decrease of $^{137}$Cs concentration

in the upper layer of sediments. Resuspension and lateral transport of the fine-grained sediments also can redistribute radiocaesium in the coastal sediment (Otosaka and Kobayashi, 2013). The simplified representation of the exchange processes in the upper layer of the sediment and the lack of re-suspension cannot account for the mechanisms described above. Instead, to take into account the vertical transfer of $^{137}$Cs we added the exchange terms $(C_{s,1} - C_{s,2})\lambda_s$ and $-(C_{s,1} - C_{s,2})\lambda_s$ to the

right hand side of the equations for the concentration of radioactivity in upper ($C_{s,1}$) and medium ($C_{s,2}$) layers of sediment in the coastal box , respectively. It can be seen that corrected prediction in Fig. 3b is in good agreement with the observations for 2012-2014. The geometric mean of the simulated-to-observed values is 0.93, with a geometric standard deviation of 1.26 for a total number of observations $N$=42 for the period 1984-2014.

The simulated $^{137}$Cs concentrations in deposit feeding invertebrates, demersal fishes, bottom predators and coastal predators in the coastal box are shown in Fig. 4 along with observed con-



centrations by the Japan Fisheries Research Agency (JFRA, 2015). As seen in Fig. 4a, just after the accident the simulated $^{137}$Cs concentration in the deposit feeding invertebrates and the observed concentration in the sea urchin increase due to the high concentration of $^{137}$Cs in the water (Fig. 4a).

After that the concentration trend becomes equal to the bottom sediment trend. The decrease constant of the fitted exponential function of simulated concentration (depuration constant) is 0.45 y$^{-1}$, which is close to the decrease constant for the sediment observations (0.44 y$^{-1}$). It agrees with the conclusion by Sohtome et al. (2014) that both observed decrease rates of concentration in sediment and in deposit-feeding benthic invertebrates are almost identical. The predicted transfer coefficient from

bulk sediment to deposit feeding benthic invertebrates for the period of 2012-2020 is approximately 0.07. Results of observations and rearing experiment for benthic polychaete (Shigenobu et al., 2015) showed that this coefficient was less than 0.1. The geometric mean of the simulated-to-observed values is 0.95, with a geometric standard deviation of 1.43 for a total number of observations $N$=20.

The results of simulation of the $^{137}$Cs concentration in the demersal fishes (Fig. 4b) agree well

with observations documented for several species of flounders. The geometric mean of the simulated-to-observed values is 1.18, with a geometric standard deviation of 1.30 for a total number of observations $N$=47. The simulated depuration rate is 0.46 y$^{-1}$ whereas the experimental value for 2012-2014 is 0.48 y$^{-1}$. The gradual decrease of activity in demersal fish caused by the transfer of activity from organic matter deposited in the bottom sediment is similar to observations by Wada et

al. (2013). Notice that the predicted transfer coefficient from bulk sediment to demersal fish for the period of 2012-2020 is approximately 0.13. This value is larger than that for deposit feeding invertebrates due to the biomagnification effect. Comparison of simulations with observations for a bottom predator (rockfish) in Fig. 4c shows a good agreement. The geometric mean of the simulated-to-observed values is 0.8, with a geometric standard deviation of 1.75 for a total number of observations

$N$=46. The comparison of simulated and observed concentrations of $^{137}$Cs in coastal predators is given in Fig. 4d. The open and filled symbols are data for seabass and fat greenling, respectively. The geometric mean of the simulated-to-observed values for coastal predators is 1.19, with a geometric standard deviation of 1.85 for a total number of observations $N$=66 for the period of 1984-2014. As seen in Fig. 4d, the simulated concentration of $^{137}$Cs in coastal predators feeding on both pelagic and

benthic organisms is similar to the simulated concentration in pelagic piscivorous fish in the period of 2011-2013, whereas after 2013 the concentration in coastal predators decreases more slowly than in piscivorous fish due to the omnivorous predation diet of coastal predators that includes benthic organisms.

The model output can be sensitive to the model parameters which are known with high uncertainty.

Therefore, a sensitivity study was carried out for the major benthic food web parameters including the water uptake rate $K_{w,i}$, the food uptake rate $K_{f,i}$, the biological half-life of $^{137}$Cs in the organism $T_{0.5}$ and for the concentration ratio of assimilated radioactivity from the organic fraction in bottom sediment to the radioactivity in bulk bottom sediment $\phi_{org}$. The effects from variations in these



parameters were estimated for the following model output: maximum $^{137}$Cs concentration in the
i=2,..11 types of organisms in the coastal box after the FDNNP accident. The range for $K_{w,i}$, $K_{f,i}$,
$T_{0.5,i}$, and $\phi_{org}$ is defined following Keum et al. (2015) as follows: minimum value is half the
reference value and maximum value is twice the reference value. The reference values for $K_{w,i}$,
$K_{f,i}$, and $T_{0.5,i}$ are given in Tables 1 and 2 whereas $\phi_{org} = 0.01$. The model output sensitivity was
estimated using sensitivity index (SI). It was calculated following Hamby (1994) as

$$SI = \frac{D_{max} - D_{min}}{D_{max}}, \tag{8}$$

where $D_{max}$ and $D_{min}$ are output values for maximal and minimal values in the parameter range,
respectively.

Figure S3a shows that all organisms (except the primary producers) are most sensitive to the vari-
ation of $K_{f,i}$. Effect of the food uptake rate for zooplankton $K_{f,2}$ slightly decreases up the pelagic
food web ($i = 2, 3$), whereas it is much less for the benthic food web ($i = 7 - 11$) because of its
diverse diet. The biological half-life for zooplankton $T_{0.5,i}$ was also one of most sensitive param-
eters both for pelagic and benthic food webs (Fig. 3b). The maximum $^{137}$Cs concentrations for
zooplankton, non-piscivorous and piscivorous fishes, algae, deposit-feeding invertebrates, molluscs,
crustaceans, demersal fish, bottom predators and coastal predators using maximal value of $T_{0.5,i}$
were increased by a factor 2.7, 2.4, 2, 1.3, 1, 1.3, 1.9,1,1.3, and 1.7, respectively, compared with a
case when a minimum values of $T_{0.5,i}$ was used. The biological half-life $T_{0.5,6}$ of deposit feeding
invertebrates essentially influences $^{137}$Cs concentration in demersal fish ($i = 9$). Figure S3c shows
that the effect of variations in the water uptake rate of zooplankton $K_{w,2}$ decreased for organisms of
higher trophic levels, showing good agreement with results by Keum et al. (2015). The concentra-
tions of $^{137}$Cs in algae and deposit-feeding invertebrates are found to be three times more sensitive
to the variations in water uptake than in the rest of organisms. The benthic organisms were less
sensitive to the parameter $\phi_{org}$ (Fig. 3d).

To estimate the contribution of benthic organisms in the individual ingestion dose rate due to
the consumption of contaminated marine products, a hypothetical reference group is considered. It
is assumed that this reference group is located in the Fukushima region and consumes only ma-
rine products from the coastal compartment near the Fukushima NPP. According to data given by
Povinec et al. (2013), the annual consumption of marine products in Japan is 23.4 kg of fish, 2 kg
of crustaceans, 1.3 kg of molluscs, and 3.7 kg of macro-algae. We compare two cases assuming
that the consumed fish are pelagial or benthic species. In the first case consumption of piscivorous
and non piscivorous species are equal. In the second case consumption of each of the three species
of fish (demersal fish, bottom predator and coastal predator) is 1/3. In both cases we consider the
period of 2014-2020 because the simulated and observed radiocaesium concentration in fish for
2011-2013 exceeds the Japanese regulatory level of seafood safety (100 Bq kg$^{-1}$) for Fukushima
offshore waters (Fig.4). In the first case the fish consumption is equally divided between piscivorous
and non piscivorous species. The dose contributions of the two caesium-isotopes $^{134}$Cs and $^{137}$Cs





for the period of 2014-2020 are 1.4 $\mu$Sv and 6.3 $\mu$Sv, respectively. In the second case the consumption of each of the three species (demersal fish, bottom predator and coastal predator) is 1/3 from 23.4 kg. The corresponding dose contributions are 29$\mu$Sv and 56 $\mu$Sv,respectively. The total dose for consumed marine products including pelagic fish (7.7 $\mu$Sv) is one order smaller than for marine

products including benthic fish (85 $\mu$Sv) but both of them are much less than the maximum annual effective dose commitment for the public, equal to 1000 $\mu$Sv according to IAEA regulations (IAEA, 2014). Notice that we considered a conservative scenario with a continuous leak of radiocaesium from FDNPP in the period of 2012-2020, whereas ending of this leak results in a return of $^{137}$Cs concentration to background value within one year (Maderich et al., 2014a).

**4    Modelling the effects from the Chernobyl accident on marine organisms in the Baltic Sea**

**4.1    Model setup**

The model was customized for the Baltic Sea, the North Sea, and the North Atlantic Ocean. The box system contains a total of 81 regional boxes followed by an additional 16 boxes for the inflow from rivers into the Baltic Sea. A plot of the box system is shown in Fig. 5. The volume and av-

erage depth for the 47 boxes describing the Baltic Sea are derived from bathymetric data. A water column with a depth of more than 60 m is divided into two layers (surface and bottom) to allow for activity stratification in the water column. These multi-layered boxes are marked blue in Fig.5. The exchange of water between the boxes in the Baltic Sea is based on a ten year average (1991-2000)of three-dimensional currents from reanalysis based on the Swedish Meteorological and Hydrologi-

cal Institute (SMHI) model (SMHI, 2013). The exchange rates for the remainder of the boxes were adopted from the standard POSEIDON configuration (Lepicard et al., 2004). To consider the water balance of the Baltic Sea and the inflow of radioactivity from river runoff, an additional 16 boxes were defined to represent main rivers in the basin. The inflow of river water for each box is based on information reported by Leppäranta and Myrberg (2009). The total inflow of water into the rivers

accumulates to 484 km$^3$y$^{-1}$. A concentration of suspended sediments (different for each box) was calculated by a 3D hydrodynamic THREETOX model (Margvelashvily et al., 1997; Maderich et al., 2008). The bottom sediment classes for simulation were determined using data from Winterhalter et al, (1981). The simulation of transport and fate of $^{137}$Cs in the Baltic Sea was carried out for the period 1945-2020. The main sources of $^{137}$Cs as included in this model are: global deposition

from weapon testing and from the Chernobyl accident (HELCOM,1995), release from the Sellafield and La Hague reprocessing plants (HELCOM, 2009), regional deposition from the Chernobyl accident in May 1986 (HELCOM, 1995), and river runoff. Details of these main sources are shown in Fig. S4a (global deposition), and in Fig. S4b (Sellafield and La Hague releases), as well as in Table S1 (Chernobyl accident). The river runoff from corresponding catchments was calculated using a

generic model by Smith et al. (2004). The value for parameter $\phi_{org}$ is 0.02.



## 4.2 Results

The simulation results for the period of 1945-2020 are shown in Fig. 6-7 for box 45 where data for concentration in the water, in the sediment and in the biota are most detailed (MARIS, 2015; MORS, 2015). Time variations of $^{137}$Cs concentration in the water and sediments in Fig. 6 show two maxima
related with weapon testing and the Chernobyl accident and then with a decreasing tendency due to outflow to the North Sea and radioactive decay. The decrease constants of the fitted exponential function of the simulated concentration in the water (0.081 y$^{-1}$) and sediments (0.070 y$^{-1}$) are close unlike the Fukushima accident where the plume of contaminated water quickly dissolves in open ocean. The simulation results are in good agreement with the measurements. The geometric
mean ratios between the predicted and observed values for concentration in the water and sediment are 0.89 and 0.86, respectively. The geometric standard deviation for concentration in the water is of 1.42 for a total number of observations $N$=378, whereas corresponding value for concentration in the sediment is of 2.17 for a total number of observations $N$=163 in the whole Baltic Sea.

    Figure 7 shows a comparison between the calculated and observed $^{137}$Cs concentration in marine
organisms for box 45. Comparison of the calculated concentrations of $^{137}$Cs in the deposit-feeding invertebrates with the measurements (Fig. 7a) shows that the model correctly predicts the time-varying concentration in these organisms. The assessment of the model accuracy in this case is, however, hardly possible because of the small number of measurements. Calculated and observed concentrations of $^{137}$Cs in pelagic non-piscivorous fish (sprat) demonstrate a good agreement with
the measurements (Fig. 7b). The geometric mean ratio for the simulated-to-observed values is 0.91 with a geometric standard deviation of 1.32 for a total number of observations $N = 24$ in the whole Baltic Sea. Using the standard model with a constant value of $CF_{ph}$ (IAEA, 2004) for brackish waters leads to an essential underestimation of the concentration in fish: the geometric mean value is 0.68 with a geometric standard deviation of 1.33. Comparison of calculated and observed con-
centrations of $^{137}$Cs in demersal fish (European flounder) is shown in Fig. 7c. It can be seen that the concentration of $^{137}$Cs in demersal fish reveals a pattern with significantly more delay in time compared with that in the non-piscivorous fish (Fig. 7b) due to the difference in the food chain between these species (Fig. 1). The benthic food web depends on the $^{137}$Cs concentration in the bottom sediment (Fig. 6b), which follows the $^{137}$Cs concentration in water with some delay (Fig. 6a). No-
tice that European flounder belongs to the polychaete and small crustacean feeding group (Gibson et al., 2015). The geometric mean ratio for the simulated-to-observed values is 0.92 with a geometric standard deviation of 1.67 for a total number of observations $N$=70 in the whole Baltic Sea. Calculated and observed $^{137}$Cs concentration in the coastal predator (cod) also agree well with the measurements (Fig. 7d): the geometric mean ratio for the simulated-to-observed values is 0.91 with
a geometric standard deviation of 1.37 for a total number of observations $N$=95 in the whole Baltic Sea. The concentration of $^{137}$Cs in the coastal predators is greater than in piscivorous fish due to the additional benthic food chain included in the web (Fig. 7d). In contrast to the open Pacific Ocean



coast where the FDNPP is located, concentrations in demersal fish, pelagic and coastal predators after the Chernobyl accident decrease with almost the same rate (about 0.075 $y^{-1}$). The variation in decrease rate is approximately 10% with a decrease rate 0.081 $y^{-1}$ for water and 0.07 $y^{-1}$ for sediment. The weak water exchange with the North Sea of the semi-enclosed Baltic Sea results in a slow evolution of water-sediments-biota system in quasi-equilibrium state. Notice that the food-web model parameters, except for the correction for brackish waters, are the same as for the FDNPP case study demonstrating generic character of the model.

## 5 Conclusions

A generic dynamic food web model was extended to include the benthic food chain. In the model pelagic organisms are grouped into phytoplankton, zooplankton, non-piscivorous fish and piscivorous fish (Heling et al., 2002). The benthic organisms are grouped into deposit feeding invertebrate, demersal fish, and bottom predators. The components of this system also include crustaceans, molluscs and coastal predators. The model takes into account the salinity effect on the intake of radiocaesium. The foodweb model is embedded into the POSEIDON-R compartment model (Lepicard et al., 2004; Maderich et al., 2014a,b) where the marine environment is modelled as a system of compartments comprising the water column, bottom sediment and biota. The compartment model was applied to two regions (north western Pacific and the Baltic Sea) that were strongly contaminated due to accidents on the Fukushima Dai-ichi and Chernobyl NPPs. Results of simulations were compared with available data for the period of 1945-2015. The modeling confirmed the presence of a continuous leakage of $^{137}$Cs from Fukushima Dai-ichi NPP with a rate of 3.6 TBq $y^{-1}$ from 2012 resulting in an almost constant concentration of $^{137}$Cs in an area of 15x30 km around the NPP. The decrease rate for the simulated concentration in the deposit ingesting invertebrates (0.45 $y^{-1}$) is close to the decrease rate for the sediment concentration (0.44 $y^{-1}$) found experimentally. This is due to a diverse diet of invertebrates, and this is conformed with the conclusions by Sohtome et al. (2014) that the decrease of observed concentration in sediment and deposit-feeding benthic invertebrates is almost identical. The predicted-by-model low (0.07) transfer coefficient of radiocaesium from bulk sediment to deposit-feeding benthic invertebrates in the area around the FDNPP for the period of 2012-2020 is consistent with observations and rearing experiments (Shigenobu et al., 2015). The findings are comparable with observations by Wada et al. (2013) showing a gradual decrease of activity in the demersal fish (decrease constant is 0.46 $y^{-1}$) caused by transfer of activity from organic matter deposited in bottom sediment through the deposit feeding invertebrates. The estimated model transfer coefficient from bulk sediment to demersal fish for the period of 2012-2020 (0.13) is larger than that for deposit feeding invertebrates due to the biomagnification effect. This value can be used for mapping of demersal fish contamination from the bottom sediments. The concentration in coastal predators that feed on both pelagic and benthic organisms is similar to the concentration in pelagic



piscivorous fish for the period of 2011-2013 when effects of water contamination were dominant. After 2013 the concentration in coastal predators decreases slower than in piscivorous fish due to the omnivorous predation diet of coastal predator that includes benthic organisms.

The total individual dose of a reference group for consumed marine products including only pelagic fish contaminated by two caesium-isotopes $^{134}$Cs and $^{137}$Cs from a coastal compartment in the period 2014-2020 is 1.4 $\mu$Sv and 6.3 $\mu$Sv, respectively. The total dose contribution for marine products including pelagic and benthic fish of the two caesium-isotopes $^{134}$Cs and $^{137}$Cs for the same period are 29 $\mu$Sv and 56 $\mu$Sv, respectively. The total dose for consumed marine products from pelagic fish (7.7 $\mu$Sv) is one order smaller than when including pelagic and benthic fishes (85 $\mu$Sv) but both of them are much less than the maximum annual effective dose commitment for the public, equal to 1000 $\mu$Sv according to IAEA regulations (IAEA, 2014).

The results of the application of POSEIDON-R with an extended dynamic model to the Baltic Sea which is semi-enclosed and filled by brackish waters are in good agreement with available measurements in the Baltic Sea. Unlike the off coast processes in the Pacific Ocean where the FDNPP is located, weak water exchange with the North Sea results in slow quasi-equilibrium evolution of water-sediments-biota system. The obtained results demonstrate the importance of the benthic food chain in the long-term transfer of $^{137}$Cs from contaminated bottom sediments to marine organisms and the potential of a generic model for use in different regions of the World Ocean.

*Acknowledgements.* This work was supported by KIOST major project (PE99304), CKJORC (China-Korea Joint Ocean Research Center) Project for Nuclear Safety, FP7-Fission-2012 project PREPARE "Innovative integrative tools and platforms to be prepared for radiological emergencies and post-accident response in Europe" and it is in frame of IAEA MODARIA Programme "Modelling and data for radiological impact assessments".



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



**Table 1.** Parameters of dynamical food chain model.

| $i$ | Organism | $drw$ | $K_{f,i}$ d$^{-1}$ | $a_i$ | $K_{w,i}$ m$^3$kg$^{-1}$d$^{-1}$ | $b_i$ | $T_{0.5,i}$ d |
|---|---|---|---|---|---|---|---|
| 1 | Phytoplankton | 0.1 | | | | | |
| 2 | Zooplankton | 0.1 | 1.0 | 0.2 | 1.5 | 0.001 | 5 |
| 3 | Non-piscivorous fish | 0.25 | 0.03 | 0.5 | 0.1 | 0.001 | Table 3 |
| 4 | Piscivorous fish | 0.3 | 0.007 | 0.7 | 0.075 | 0.001 | Table 3 |
| 5 | Macroalgae | 0.1 | | | 0.6 | 0.001 | 60 |
| 6 | Deposit feeding invertebrate | 0.1 | 0.02 | 0.3 | 0.1 | 0.001 | 15 |
| 7 | Mollusc | 0.1 | 0.06 | 0.5 | 0.15 | 0.001 | 50 |
| 8 | Crustacean | 0.1 | 0.015 | 0.5 | 0.1 | 0.001 | 100 |
| 9 | Demersal fish | 0.25 | 0.007 | 0.5 | 0.05 | 0.001 | Table 3 |
| 10 | Bottom predator | 0.3 | 0.007 | 0.7 | 0.05 | 0.001 | Table 3 |
| 11 | Coastal predator | 0.3 | 0.007 | 0.7 | 0.075 | 0.001 | Table 3 |





**Table 2.** Preference of predator of type $i$ for prey of type $j$.

| Predator<br>Prey | 2 | 3 | 4 | 6 | 7 | 8 | 9 | 10 | 11 |
|---|---|---|---|---|---|---|---|---|---|
| 0 |  |  |  | 0.5 |  |  | 0.1 |  |  |
| 1 | 1.0 |  |  |  | 0.6 | 0.1 |  |  |  |
| 2 |  | 1.0 |  |  | 0.2 | 0.8 |  |  |  |
| 3 |  |  | 1.0 |  |  |  |  |  | 0.2 |
| 5 |  |  |  | 0.5 | 0.2 | 0.1 |  |  |  |
| 6 |  |  |  |  |  |  | 0.7 | 0.3 | 0.25 |
| 7 |  |  |  |  |  |  | 0.1 | 0.2 | 0.1 |
| 8 |  |  |  |  |  |  | 0.1 | 0.2 | 0.2 |
| 9 |  |  |  |  |  |  |  | 0.3 | 0.25 |





**Table 3.** Parameters for the fish in dynamical food chain model.

| Target tissue | Bone | Flesh | Organs | Stomach |
|---|---|---|---|---|
| Weght fraction | 0.12 | 0.80 | 0.05 | 0.03 |
| Target tissue modifier | 0.5 | 1.0 | 0.5 | 0.5 |
| Biological half-life of non-piscivorous fish (d) | 500 | 75 | 20 | 3 |
| Biological half-life of piscivorous fish (d) | 1000 | 150 | 40 | 5 |
| Biological half-life of demersal fish (d) | 500 | 75 | 20 | 3 |
| Biological half-life of bottom predator fish (d) | 1000 | 150 | 40 | 5 |
| Biological half-life of coastal predator fish (d) fish (d) | 1000 | 150 | 40 | 5 |





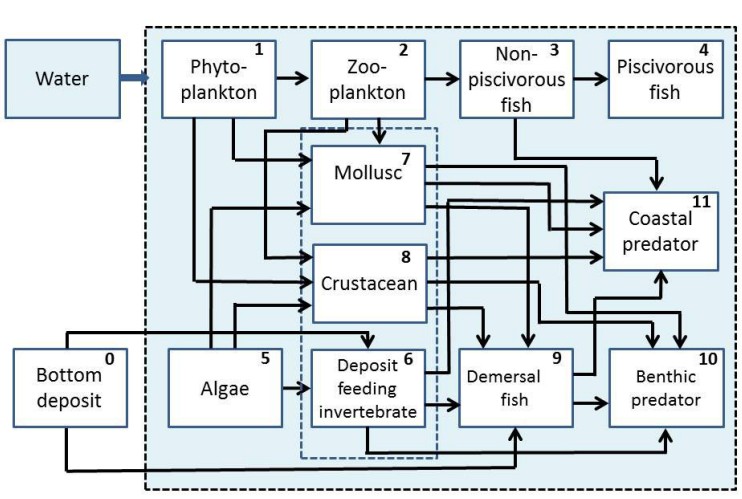

**Figure 1.** Scheme of radionuclide transfer to marine organisms.





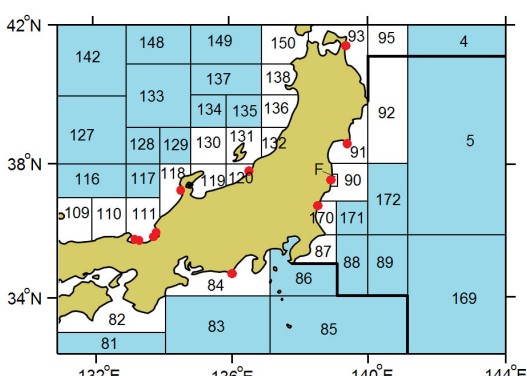

**Figure 2.** The box system for the area close to Fukushima NPP. The shaded boxes represent the deep water boxes. The NPPs are shown by filled circles. Coastal box around the FDNPP (marked by "F" is inside of box 90. Thick line limits the area of the Fukushima accident fallout.



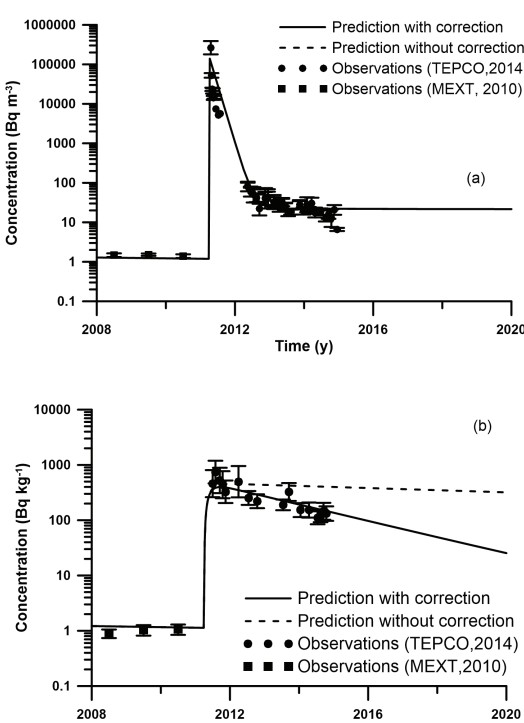

**Figure 3.** Comparison between calculated and observed $^{137}$Cs concentration in seawater (a) and in bottom sediment (b) in the coastal box around the Fukushima Dai-ichi NPP.





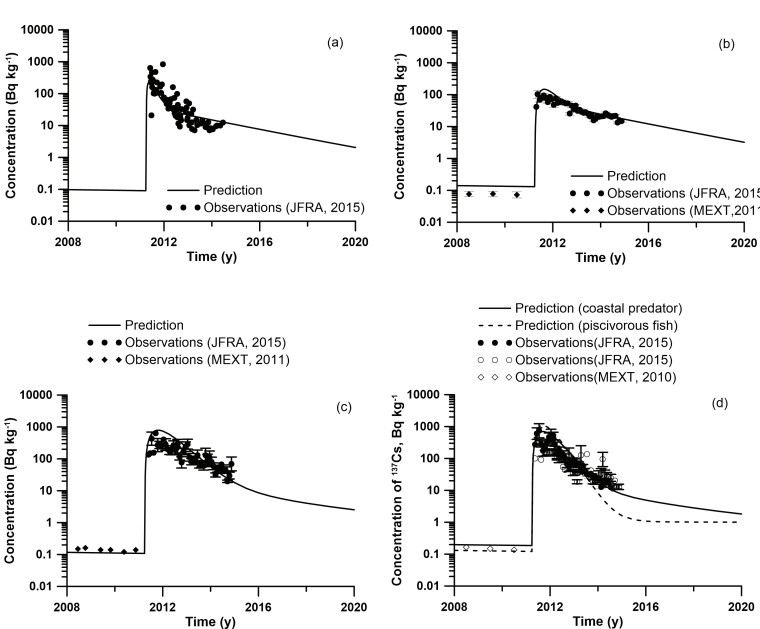

**Figure 4.** Comparison between calculated and observed $^{137}$Cs concentration in deposit feeding invertebrate (a), demersal fish (b), bottom predator (c) and coastal predator (d) around the FDNPP.





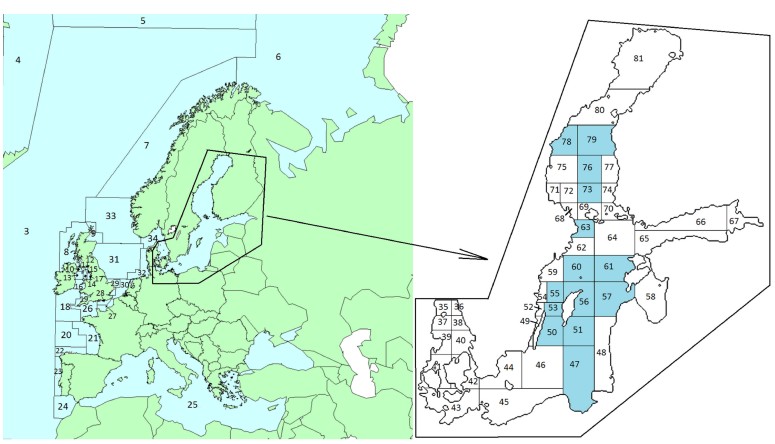

**Figure 5.** Compartment system of POSEIDON-R model for north-eastern part of Atlantic Ocean, the North Sea and the Baltic Sea.





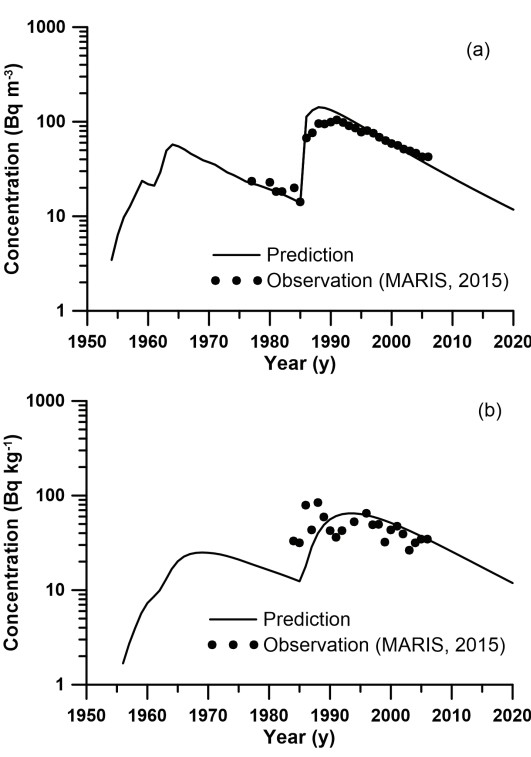

**Figure 6.** Comparison between calculated and observed $^{137}$Cs concentrations in seawater (a) and in bottom sediment (b) for box 45.



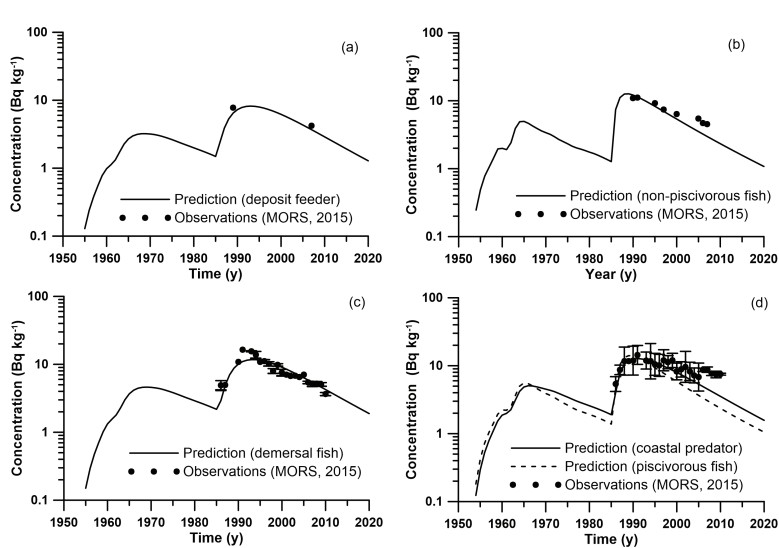

**Figure 7.** Comparison between calculated and observed $^{137}$Cs concentrations in deposit-feeding invertebrate (a), non-piscivorous fish (b), demersal fish (c) and coastal predator (d) for box 45.