# Peer review of "TRANSFER OF RADIOCAESIUM FROM CONTAMINATED BOTTOM SEDIMENTS TO MARINE ORGANISMS THROUGH BENTHIC FOOD CHAIN IN POST-FUKUSHIMA AND POST-CHERNOBYL PERIODS"

_Biogeosciences, 2015_

## Referee Comment (RC1) · Anonymous Referee #1 · 24 Feb 2016

Scientific significance: This is an inspiring paper which includes a dynamic pelagic food chain as well as benthic food chain. This is seldom seen in radioecology and hopefully opens up a world of new ideas in radioecology. At the same time it needs to maintain its connections to marine ecology, where the benthic and pelagic food webs have been studied for a long time. Thus it needs to be understandable both for radioecologist and marine ecologist, which can be difficult to achieve. Below are some comments how this can be improved. The presentation quality of paper is good, well written and

structured, even if the connection between the two sites seem to be only the model and Cs. I don't see any discussion or comparison what the difference is between the sites, just examples. The scientific quality has a good appearance, but when looking closer to the supporting model and references the results are weak. There is simple to little data to support the modelling results (exemplified below) . Moreover the scientific nomenclature is not consistent with e.g. marine ecological nomenclature and exact description of e.g. species.

A PDF file attached with detailed comments

Please also note the supplement to this comment:
http://www.biogeosciences-discuss.net/bg-2015-654/bg-2015-654-RC1-supplement.pdf

**Supplement:**

Scientific significance: This is an inspiring paper which includes a dynamic pelagic  food chain as well as benthic food chain. This is seldom seen in radioecology and hopefully opens up a world of new ideas in radioecology. At the same time it needs to maintain its connections to marine ecology, where the benthic and pelagic food webs have been studied for a long time. Thus it needs to be understandable both for radioecologist and marine ecologist, which can be difficult to achieve. Below are some comments how this can be improved.

The presentation quality of paper is good, well written and structured, even if the connection between the two sites seem to be only the model and Cs. I don't see any discussion or comparison what the difference is between the sites, just examples.

The scientific quality has a good appearance, but when looking closer to the supporting model and references the results are weak. There is simple to little data to support the modelling results (exemplified below) .  Moreover the scientific nomenclature is not consistent with e.g. marine ecological nomenclature and exact description of e.g. species.

1. Although it is a step forward to include the foodchains, both the bentic and pelagic foodweb presented here miss the important microbial loops and even meiofauna. The microbial loop has been discussed the last three decades in oceanography and limnology ( see review in Fenchel  2008  Journal of Experimental Marine Biology and Ecology 366 (2008) 99–103 . The depicted foodchain in figure 1 doesn't include and nothing is mentioned in the text. I can understand that there reasons not to include them in the model, but there a no reasons to omit them without explanation.  This will certainly cause doubts of sound science of this paper by marine ecologist on this paper.
2. There are problems with the classifications in the paper of the different trophic  groups discussed in the paper and shown in figure 1. They are not consistent and classification with the same variables.  E.g what is a costal predator and what difference compared to piscivoures fish? The example given is cod. In the Baltic Sea it certainly would be regarded a Piscivorus fish you can everywhere not only coast.  Algae (fig 1) and phytoplankton are the same, you maybe mean benthic algae or macroalgae. In table 1 you call it macroalgae Demersal fish and Benthic predator what is the difference? Example is given with European flounder which certainly is a benthic predator and demersal fish at the same time. In Fukushima we can read about Rockfish , what is that? There are least a dozen fish genera which can be called rockfish, they have different position in the food chain. This needs better description or at least the scientific ( latin) name.
3. There are other filterfeeders than mollusc and mollusc can be grazers and deposit feeders. Benthic algae are consumed by grazer also. Why the difference between deposit feeding invertebrates and crustacean invertebrates Crustaceans are certainly many of the Zooplankton.
4. No explantion in figure 1 what are the arrows, boxes, dotted lines, where are the explantions of the categories. What are the numbers?The dotted box with a waterbox  outside? Water deposit what is that and why is that box outside ?
5. Figure 2. What are deep water boxes ? What are coastal box? Describe or give citeria or point to text where that is described

6. Figure 3 Explain what the legend means e.g correction of wha?t . kg of what drymatter ?? Something strange that the estimated KD for the sediment is different before and after Fukushima, especially the before values the ratio seem low if you expect a KD of 1000 l/kg

7. The paper identifies that an important process is missing resuspension, and it is compensation with some unclear equations. In the Baltic Sea this certainly is a much more important process than e.g. diffusion. I can imagine that it could be important outside Fukushima especially since the organic content is so low (<25% line 50). Thus it would certainly lift this to a through discussion and conclusion, not just an equation fix.

8. Line 26 : What does the biomagnification effect mean here? See e.g. Gray 2002 Marine Pollution Bulletin 45 (2002) 46–52 Biomagnification in marine systems: the perspective of an ecologist

9. Line 50 says that it is bound to organic matter. It is unclear if that means dead matter och e.g microbes or both.

10. Line 66: What is an underground leakage in this context?

11. Line 109: I don't understand "rapid and more intensive transfer of several sediment adsorbed radionuclides to particular organs of the demersal fishes" in contrast invertebrates? Or as another source of contaminants?

12. Line 111: In this context I don't understand the role of macroalgae (and why not benthic microalgae) I would also assume that crustaceans and molluscs are able to graze the algae not only deposit feed. Moreover there no data about the macralgae and for me if the depth of the coastal box is 60m there must be large areas outside the photic zone. How is that estimated?

13. Line 130: Why not call the food abstraction coefficient assimilation efficiency which is the normal biological word

14. Lines around 155: Since the classification not very systematic the relationships between these fish types are unclear also.

15. Line 191 and forward: The description of the POSIEIDN model should be helped with an figure showing the compartments and where the additional food web interact with POSEIDON. Moreover the parametervalue for the two sites should be tabulated somewhere, without this information it is not possible to reproduce the results.

16. Line 204: An important transfer from sediment to water column is resuspension.

17. Line 210: shallow one layer compartment ? another sedimentlayer or a description of the sectors in POSEIDON? If the later wasn't that the same compartment as the pelagic food web?

18. Line 235: Somewhere I am missing a table giving the parameters of the model. Also a description of the average depth of the site and bottom substrate is missing

19. Line 275: Do you mean the geometric mean of the ratio? Between measured to observed values?

20. Line 293: …. "cannot account"… you mean maybe " not included in the model", it is not clear.

21. Line 294: it is not easy to understand where these terms are added into which equation. This addition seem crucial to the model and needs to be presented clearer and completely to be transparent. Moreover I get the impression that this is some sort of calibration to make the model fit for the measurements. Or how is it obtained?

22. Line 304: You mention sea urchin here, is that detritus feeding or a grazer in real life and what group is it represented in the model, invertebrate?

23. Line 306: Here you mention depuration constant for the first time. I am unsure if it can be called depuration constant at least from the ectoxicological viewpoint, moreover this constant could be mentioned the first time it occurs and explained what it is.

24. Line 309: What do you mean with transfer coefficient here ? Concentration ratio?

25. Line 311: What kind of polychaete, deposit feeding or filterfeeding, to unspecific without species name

26. Line 318: Reference needed for the experimental value or/and description of the experiment. Crucial is how they are fed and how the radionuclide is added

27. Line 322: What do you mean with biomagification effect (see earlier comment) and how should that affect the CR in demersal fish mechanistically?

28. Line 323: Again inexact species and categorisation, there are several genus called rockfish, what is the scientific name? How does it fit into the classification? Coastal predator?

29. Line 326: The legend to the figure could be put into figure text

30. Line 327-333: This is interesting results and probably support your approach, but it is messed up with inconsistent classifications. My suggestion that you first of all make a consistent classification, a clear description what that means and finally give examples of species in the area for each group. This should be done in methods, the you adhere to the classification when you mention different species with common and scientific names (latin) . I know that it can never become clearcut where different species belong, but you tell at least the reader where they are in the model.

31. Line 334: ".. which are known with high uncertainity." Maybe better … known to have a high…

32. Line 352: probably figure S3b not 3b

33. Line 355: why this different numbers ?

34. Line 363: also probable wrong figure number

35. Line 363-384: I would suggest to omit this part, there are assumptions and limits with different relevance in different parts of the world. From my horizon (responsible for dose assessments) I cannot see the point of this section. Omit it the aslo from conclusion

36. Line 385-410: If the modelling of the Baltic sea should be useful, this section should at least tabulate the drivers (fluxes over borders) and parameter values for the modelled box for the result. It is not reproducible with the current information. I am also missing general data on the bathymetry and which species are considered in the model foodweb.

37. Line 440: Polychaete feeding is that valid for the Baltic Sea

38. Line 459: As commented earlier the classification system needs to be reworked

39. Line 464: Suggest to omit "strongly", it is not relevant for the Baltic Sea and I don't think I adds something more for Fukushima area.

40. Conclusion or discussion I am missing a more rigorous comparision between the Baltic Sea and Fukushima, otherwise I don't see the point include both in this paper.

41. No explanation in figure 1 what are the arrows, boxes, dotted lines, where are the explanations of the categories. What are the numbers? The dotted box with a waterbox outside? Water deposit what is that and why is that box outside ?

42. Figure 2. What are deep water boxes ? What are coastal box? Describe or give criteria or point to text where that is described

43. Figure 3 Explain what the legend means e.g correction of wha?t . kg of what drymatter ??
44. Figure 5 explain color-coding
45. Figure 6 is the concentration in bottom sediment for the bul sediment or organic fraction (the same question applies for Fukushima)
46. FigS3 There no figtext for d)

---

## Referee Comment (RC2) · Anonymous Referee #2 · 18 Mar 2016

The paper is substantially corrected which makes reader more being easier to understand the result of the paper. Following points are recommended to reconsider or revise before publishing. 1) Citation of Matsumoto et al., 2015 should not be cited. The methodology in this paper (gross counts of Cs-137 energy band in whole body fish minus background counts without fish) contains bottom-up effect on net count by neglecting Compton effect from K-40 in fish flesh, which makes the turnover rate being overestimated. 2) Rational for radiocesium turnover in bone etc. other than flesh

should be shown by citation or theoretical assumption. It will help following further research by similar approach.

Please also note the supplement to this comment:
http://www.biogeosciences-discuss.net/bg-2015-654/bg-2015-654-RC2-supplement.pdf

―――――――――――――――――

**Supplement:**

[revised manuscript text omitted]

---

## Author Response (AR1)

**Response to Reviewer #1**

The authors are most grateful to the reviewer for thorough analysis of manuscript and for his constructive criticism and suggestions. We have taken his remarks into account, and the paper has been revised in many places accordingly.

*Scientific significance: This is an inspiring paper which includes a dynamic pelagic food chain as well as benthic food chain. This is seldom seen in radioecology and hopefully opens up a world of new ideas in radioecology. At the same time it needs to maintain its connections to marine ecology, where the benthic and pelagic food webs have been studied for a long time. Thus it needs to be understandable both for radioecologist and marine ecologist, which can be difficult to achieve. Below are some comments how this can be improved.*
*The presentation quality of paper is good, well written and structured, even if the connection between the two sites seem to be only the model and Cs. I don't see any discussion or comparison what the difference is between the sites, just examples.*
*The scientific quality has a good appearance, but when looking closer to the supporting model and references the results are weak. There is simple to little data to support the modelling results (exemplified below) . Moreover the scientific nomenclature is not consistent with e.g. marine ecological nomenclature and exact description of e.g. species.*

**Answer.** The model results were compared with observations in two very different marine environments: in the North Western Pacific and in the Baltic Sea before and after Fukushima and Chernobyl accidents, respectively. The added observations for 2015-2016 in Figs. 3 and 4 support the generic model predictions. These figures and updated Supplement are given after text of response. The detailed answers on the rest of reviewer comments are given below.

  1. *Although it is a step forward to include the foodchains, both the bentic and pelagic foodweb presented here miss the important microbial loops and even meiofauna. The microbial loop has been discussed the last three decades in oceanography and limnology ( see review in Fenchel 2008 Journal of Experimental Marine Biology and Ecology 366 (2008) 99–103 . The depicted foodchain in figure 1 doesn't include and nothing is mentioned in the text. I can understand that there reasons not to include them in the model, but there a no reasons to omit them without explanation. This will certainly cause doubts of sound science of this paper by marine ecologist on this paper.*

**Answer:** We agree that a full model of pelagic and benthic food webs should include a variety of transfer processes in water and in the sediment. However, we consider here the more limited task of biota model development and its implementation into the compartment model which is in turn a component of the decision-support system RODOS for nuclear emergency. Therefore, a number of simplifications have been made in order that the model is robust and generic, requiring a minimum number of parameters. It is assumed that the radioactivity concentrations in organic and mineral fractions of bottom deposit are in mutual equilibrium, and the radioactivity concentrations in microbial biota and non-living organic matters also are in equilibrium, and only organic matter in the bottom deposit is bioavailable. The text was changed accordingly (see answer on comment #2). To explain model assumptions and limitations we reworked text in lines 94-99 as

"To describe transfer pathways of $^{137}$Cs from bottom sediments to marine organisms the dynamic model BURN was extended. The model was developed to assess doses from marine products in the decision-support system RODOS for off-site nuclear emergencies (Lepicard et al., 2004). For such aim it was necessary to use a robust and generic model requiring a minimal number of parameters. Therefore, in the model the marine organisms are grouped into a few classes based on trophic levels and types of species. The radionuclides are also grouped in several classes in terms of tissues in which a specific radionuclide accumulates preferentially. These simplifications allow for a limited number of standard input parameters."

2. *There are problems with the classifications in the paper of the different trophic groups discussed in the paper and shown in figure 1. They are not consistent and classification with the same variables. E.g what is a costal predator and what difference compared to piscivoures fish? The example given is cod. In the Baltic Sea it certainly would be regarded a Piscivorus fish you can everywhere not only coast. Algae (fig 1) and phytoplankton are the same, you maybe mean benthic algae or macroalgae. In table 1 you call it macroalgae Demersal fish and Benthic predator what is the difference? Example is given with European flounder which certainly is a benthic predator and demersal fish at the same time. In Fukushima we can read about Rockfish , what is that? There are least a dozen fish genera which can be called rockfish, they have different position in the food chain. This needs better description or at least the scientific ( latin) name.*

**Answer:** We agree with referee's comment #30 "*that it can never become clearcut where different species belong.*" A good example is the omnivorous predator Atlantic cod (Gadus morhua) in the Baltic Sea. Diet of cod in deep Central Baltic can be dominated by herring and sprat. However in shallow Western Baltic (box 45 in Fig. 5, depth 31.4 m) diet is diverse, including herring, sprat, Gobiidae, the molluscs, various Polychaeta and crustaceans (Sparholt, 1994). Therefore for this basin the cod is considered as "coastal predator" feeding by both pelagic and benthic preys. Following reviewer's comment, "Algae" in Fig.1 renamed to "Macroalgae". According Gibson and al. (2015) European flounder (Platichthys flesus) belongs to group of "Polychaete and small crustacean feeders". See more details in answer on comment #37. Rockfish is "Japanese rockfish" (Sebastes Cheni). The Latin names are given for species when observations and simulation results are compared (see answers on comments #28 and #36).

The text and caption to Fig. 1 were changed to extend description of food web and explain classification approaches used in the paper.

Lines 99-114 "The transfer scheme of radionuclides through the marine food web is shown in Fig. 1 where transfer of radionuclides through the food web is shown by arrows whereas the direct transfer from water is depicted by the shadowed rectangle surrounding 11 biota compartments ($i$=1,…,11). The different food-chains exist in both pelagic and benthic zones. Pelagic organisms are divided into primary producer, phytoplankton ($i$=1), and consumers which consist of zooplankton ($i$=2), forage (non-piscivorous) fish ($i$=3), and piscivorous fish ($i$=4). The benthic food web includes three primary pathways for radionuclides: (i) transfer from water to macroalgae ($i$=5), then to grazing invertebrates ($i$=6,..,8); (ii) through the vertical detritus flux and zooplankton faeces (Fowler et al., 1987) to detritus-feeding invertebrates, and (iii) through contaminated bottom sediments to deposit feeding invertebrates. Concentrations of radionuclides in water and in the upper layer of bottom sediment are calculated using the box model POSEIDON-R described below. The output

from this model is shown by external boxes in Fig. 1. The radionuclides adsorbed on the organic matter in the sediments are bioavailable for benthic organisms but the mineral component of sediments is not (Ueda et al., 1977; Ueda et al., 1978) although Koyanagi et al. (1978) found relatively rapid and more intensive transfer of several sediment adsorbed radionuclides ($^{54}$Mn, $^{60}$Co, $^{65}$Zn) to particular organs of the demersal fishes in contrast to flesh. It is assumed that (i) radioactivity concentrations in organic and mineral fractions of bottom deposit are in mutual equilibrium, (ii) that radioactivity concentrations in microbial biota and non-living organic matter also are in equilibrium and (iii) that only organic matter in the bottom deposit is bioavailable. The benthic invertebrate group (surrounded in Fig. 1 by dashed rectangle) includes molluscs (e.g. filter—feeders) ($i$=7), crustaceans (e.g. detritus-feeders) ($i$=6) and subsurface and surface deposit feeders (e.g. annelid). It is assumed that radioactivity is transferred from invertebrates to benthic invertebrate feeding demersal fishes ($i$=9), and on to omnivorous bottom predators ($i$=10) (Fig. 1). The marine food web also includes "coastal predators" ($i$=11) feeding in the whole water column in shallow waters."

Line 443. "Calculated and observed $^{137}$Cs concentrations in the coastal predator (cod) also agree well with the measurements (Fig. 7d). The diet of Atlantic cod in shallow Western Baltic is diverse, including herring, sprat, Gobiidae, molluscs, various Polychaeta and crustaceans (Sparholt, 1994). Therefore for this basin the cod is considered as "coastal predator" feeding by both pelagic and benthic preys. The geometric mean of the simulated-to-observed ratios is 0.91 with a geometric standard deviation of 1.37 for a total number of observations $N$=95 in the whole Baltic Sea."

Sparholt, H.: Fish species interactions in the Baltic Sea. Dana, 10, 131-162, 1994.

Caption to Fig. 1
"Figure 1. Scheme of radionuclide transfer to marine organisms. A transfer of radionuclides through food web is shown by arrows whereas direct transfer from water is depicted by shadowed rectangle surrounding biota compartments. The output from the compartment POSEIDON-R model is shown by external boxes."

3. *There are other filterfeeders than mollusc and mollusc can be grazers and deposit feeders. Benthic algae are consumed by grazer also. Why the difference between deposit feeding invertebrates and crustacean invertebrates Crustaceans are certainly many of the Zooplankton.*

**Answer:** As shown in Fig. 1 and in Table 2 benthic algae are part of diet of crustaceans, molluscs and deposit-feeding invertebrates (e.g. echinoidea). Deposit feeders include subsurface deposit feeders (e.g. worms). We consider "crustaceans" as a part of benthic food web.

4. *No explantion in figure 1 what are the arrows, boxes, dotted lines, where are the explantions of the categories. What are the numbers?The dotted box with a waterbox outside? Water deposit what is that and why is that box outside ?*

**Answer:** See answer on Comment #2.

5. *Figure 2. What are deep water boxes ? What are coastal box? Describe or give criteria or point to text where that is described*

**Answer:** The text and figure caption have been changed accordingly.

Line 228 "The model was customized for the Northwestern Pacific Ocean, the East China and Yellow Seas and the East/Japan Sea. A total of 176 boxes cover this entire region (Fig. S1). In the deep-sea regions a three-layer box system was built to describe the vertical structure of the radioactivity transport in the upper layer (0-200 m), intermediate layer (200-1000 m) and lower layer (>1000m). The compartments around the FDNPP are shown in Fig. 2. The "coastal" box 15x30 km is nested into large "regional" box 90 to provide more detailed description in the area around the FDNPP. It covers observation data within a circular-shaped surface area of a radius 15 km with a center at the FDNPP. This box has one vertical layer for the water column and three bottom sediment layers. The depth of coastal box is less than that in the one layer outer box 90. The water exchange fluxes with the outer box are equal in both directions. The parameters of the coastal box are given in Table S1. The averaged advective and diffusive water fluxes between regional compartments were calculated for a ten-year period (2000-2009) using the Regional Ocean Modeling System (ROMS). Details of customization are given by Maderich et al. (2014a,b). The values for parameters $\phi_{org}$=0.01and $T_{migr,i}$=0.7 y for $i$=3,4,9,10,11 were used."

Caption to Figure 2. "The box system for the area close to Fukushima NPP. The shaded boxes represent the deep-sea water boxes divided on three vertical layers. The NPPs are shown by filled circles. Coastal box around the FDNPP (marked by "F" is inside of box 90. Thick line limits the area of the Fukushima accident fallout ."

Caption to Figure 2S "The compartment system for the Northwestern Pacific. The shaded boxes represent the deep-sea water boxes divided on three vertical layers..."

6. *Figure 3 Explain what the legend means e.g correction of wha?t . kg of what drymatter ?? Something strange that the estimated KD for the sediment is different before and after Fukushima, especially the before values the ratio seem low if you expect a KD of 1000 l/kg*

**Answer:** The "dry" (weight) was added in axis title in Figs. 3b and 6b. The $K_d$ in the simulation is constant in time. The value of $K_d$ is given in Table S1. The caption was changed accordingly.

"Figure 3. Comparison between calculated and observed [137]Cs concentrations in seawater (a) and in bottom sediment (b) in the coastal box around the Fukushima Dai-ichi NPP. The dashed line in (b) shows results of simulations using standard POSEIDON-R model, whereas solid line presents simulation with correction term in equation (S3)."

7. *The paper identifies that an important process is missing resuspension, and it is compensation with some unclear equations. In the Baltic Sea this certainly is a much more important process than e.g. diffusion. I can imagine that it could be important outside Fukushima especially since the organic content is so low (<25% line 50). Thus it would certainly lift this to a through discussion and conclusion, not just an equation fix.*

**Answer:** We used "standard" parameterization of transfer between water and bottom sediments following approach developed in series of EC MARINA projects. In this approach resuspension was not included, however, "standard" parameterization was successfully used e.g. for the Baltic Sea (MARINA-BALT). The Chernobyl case simulation confirms that the standard parameterization describes well exchange processes for the Baltic Sea (see answers on Comment #40). In the Fukushima case study we identified that [137]Cs decreases in upper layer of sediments faster than model predicts using standard parameterization. A several

possible mechanisms were mentioned in Lines 286-291 but there have been no study confirming dominance of one of these mechanisms. Therefore, the very simple parameterization was used in the model because the main aim of our study is transfer radiocaesium through the benthic food chain. The text was changed accordingly.

Line 468 "It was found that $^{137}$Cs decreases in upper layer of sediments in the Fukushima case study faster than POSEIDON-R predicts using the standard for marine compartment model parameterization of exchange between water and sediment by diffusion mechanism. A simple parameterization calibrated on measurements was therefore used to correct this exchange. However, the further studies of exchange mechanisms are necessary."

   8. *Line 26 : What does the biomagnification effect mean here? See e.g. Gray 2002 Marine Pollution Bulletin 45 (2002) 46–52 Biomagnification in marine systems: the perspective of an ecologist*
**Answer:** See answer on Comment 27

   9. *Line 50 says that it is bound to organic matter. It is unclear if that means dead matter och e.g microbes or both.*
**Answer:** There is no explicit discussion on the origin of organic matter in these papers. However, it can be concluded from the description of methods in (Ono et al., 2014) that the total organic matter passed through the 2 mm mesh sieve was tested. See also answer on comments #1 and 2.

   10. *Line 66: What is an underground leakage in this context?*
**Answer:** The routes of radioactive water from the FDNPP were not exactly identified yet (Kanda, 2013). It can be assumed that possible pathway is transport by ground water leaked from damaged facilities. Text was changed accordingly

Line 66 "In that study the flux of radionuclides due to the ground water leakage of contaminated waters from FDNPP (Kanda, 2013) was taken into account."

   11. *Line 109: I don't understand "rapid and more intensive transfer of several sediment adsorbed radionuclides to particular organs of the demersal fishes" in contrast invertebrates? Or as another source of contaminants?*
**Answer:** The text was changed:

Line 109 "although Koyanagi et al. (1978) found relatively rapid and more intensive transfer of several sediment adsorbed radionuclides ($^{54}$Mn, $^{60}$Co, $^{65}$Zn) to particular organs of the demersal fishes in contrast to flesh."

   12. *Line 111: In this context I don't understand the role of macroalgae (and why not benthic microalgae) I would also assume that crustaceans and molluscs are able to graze the algae not only deposit feed. Moreover there no data about the macralgae and for me if the depth of the coastal box is 60m there must be large areas outside the photic zone. How is that estimated?*
**Answer:** The macroalgae were considered in the food chain because they are a component in the diet of the molluscs, crustaceans and invertebrates with dominant deposit feeding (Table 2). They also are part of human diet and are important for dose estimates. We used a simple approach where the benthic component with macroalgae was included in the shallow one-layer compartments adjacent to the shore that guaranteed range of depth for macroalgae photic zone. The text was changed accordingly:

Line 209 "The model for the pelagic food web component was implemented for the upper water layer of all compartments, whereas the benthic component was included in the shallow one-layer compartments adjacent to the shore".

13. *Line 130: Why not call the food abstraction coefficient assimilation efficiency which is the normal biological word*

**Answer:** The extraction coefficient was changed on "assimilation efficiency" in Line 130

14. *Lines around 155: Since the classification not very systematic the relationships between these fish types are unclear also.*

**Answer:** See answers on comment #2.

15. *Line 191 and forward: The description of the POSIEIDN model should be helped with a figure showing the compartments and where the additional food web interact with POSEIDON. Moreover the parameter value for the two sites should be tabulated somewhere, without this information it is not possible to reproduce the results.*

**Answer:** The POSEIDON model equations and figure with compartment structure (Fig. S1) are given in Supplement. The parameters of two boxes from the Pacific Ocean and the Baltic Sea are given in Table S1. Text was changed accordingly:

Line 193 " The compartments describing the water column containing suspended matter are subdivided into a number of vertical layers as shown in Fig. S1."
Line 208 "The model equations are given in Supplement".
Line 234 "The parameters of the coastal box are given in Table S1".

16. *Line 204: An important transfer from sediment to water column is resuspension.*

**Answer:** See answer on Comment #7 .

17. *Line 210: shallow one layer compartment ? another sedimentlayer or a description of the sectors in POSEIDON? If the later wasn't that the same compartment as the pelagic food web?*

**Answer:** We described a structure of compartments in the Supplement and in the text. See answers on Comment #5. The shallow one water column layer and three sediment layer compartments include both pelagic and benthic food webs.  The text was refined as:

Line 209 "The model for the pelagic food web component was implemented for the upper water layer of all compartments, whereas the benthic component was included in the shallow one-layer compartments adjacent to the shore".

18. *Line 235: Somewhere I am missing a table giving the parameters of the model. Also a description of the average depth of the site and bottom substrate is missing*

**Answer:** We added Table S1 where these parameters were given.

19. *Line 275: Do you mean the geometric mean of the ratio? Between measured to observed values?*

**Answer**: It is geometric mean of ratio between simulated by model and observed in ocean values. The text was changed in several places as

Lines 298, 312, 327, 420, 433, 441,444   "…geometric mean of the simulated-to-observed ratios.."

*20. Line 293: …. "cannot account"… you mean maybe " not included in the model", it is not clear.*
**Answer:** The text was changed accordingly.

Line 291 "Only several of these mechanisms are included in the POSEIDON-R model."

*21. Line 294: it is not easy to understand where these terms are added into which equation. This addition seem crucial to the model and needs to be presented clearer and completely to be transparent. Moreover I get the impression that this is some sort of calibration to make the model fit for the measurements. Or how is it obtained?*
**Answer:** The text was changed accordingly.

Line 293 "Therefore, to take into account the vertical transfer of $^{137}$Cs we added the exchange terms $(C_{s,1}-C_{s,2})\lambda_s$ and $-(C_{s,1}-C_{s,2})\lambda_s$ to the right hand side of the equations (S3) and (S4) for the concentration of radioactivity in upper ($C_{s,1}$) and medium ($C_{s,2}$) layers of sediment in the coastal box, respectively. Here $\lambda_s$ is an empirical parameter. The value of $\lambda_s=0.4$ y$^{-1}$ was obtained to fit observation data for $C_{s,1}$. As seen in Fig. 3b the corrected by additional exchange term concentration of $^{137}$Cs is described well in period 2008-2015."

*22. Line 304: You mention sea urchin here, is that detritus feeding or a grazer in real life and what group is it represented in the model, invertebrate?*
**Answer**: According to Lawrence (2007) the principal foods of sea urchin (*Strongylocentrotus nudus*) include large and small algae, detritus, sand, shells, sessile animals and fish. The model diet for deposit feeding invertebrates includes both macroalgae and organic matter in the bottom deposit that grossly represent transfer of $^{137}$Cs through food to S. nudus. Notice that among of benthic invertebrates only data on the sea urchin were available for the 15 km area around the FDNPP. The text was added:

Line 305 "This is consistent with model diet that includes macroalgae and deposit organic matter grossly representing diet of S. nudus (Lawrence (2007). The macroalgae contribution in food contamination first prevails, then after 2012 the bottom contamination dominates."

Lawrence J. M. (ed): Edible sea urchins: Biology and ecology. Developments in Aquaculture and Fisheries Science, 37, Elsevier, Amsterdam, Netherlands, 529 pp., 2007.

*23. Line 306: Here you mention depuration constant for the first time. I am unsure if it can be called depuration constant at least from the ecotoxicological viewpoint, moreover this constant could be mentioned the first time it occurs and explained what it is.*
**Answer:** We defined depuration constant as a decrease constant in the fitted exponential function of concentration (see Line 305). The depuration constant is equal to $(\ln2T_{e1/2})^{-1}$, where $T_{e1/2}$ is ecological half-life. The term "depuration rate constant" is used in marine radioecology (see e.g. Sohtome et al., 2014; Tateda et al., 2013;2015).

*24. Line 309: What do you mean with transfer coefficient here ? Concentration ratio?*
**Answer:** See answer on Comment #25

*25. Line 311: What kind of polychaete, deposit feeding or filterfeeding, to unspecific without species name*
**Answer:** The text was changed accordingly comments #24-25:

Line 311. "The field studies of several species of polychaeta (deposit or filter feeders: *Flabelligeridae*, *Terebellidae* and *Opheliidae*; herbivore or carnivore feeders: *Glyceridae*, *Eunicidae*, and *Polynoidae*) off the coast of Fukushima and rearing experiment for *Perinereis aibuhitensis* demonstrated that $^{137}$Cs concentration in all specimens was much lower than that in the sediment (Shigenobu et al., 2015). Results of rearing experiment using contaminated sediments from near the FDNPP showed that transfer coefficient (concentration ratio between *P. aibuhitensis* (Bq kg$^{-1}$-wet) and contaminated sediment (Bq kg$^{-1}$-wet)) was less than 0.1. "

> 26. *Line 318: Reference needed for the experimental value or/and description of the experiment. Crucial is how they are fed and how the radionuclide is added*

**Answer:** The data are the field data from TEPCO (2015). The text was corrected accordingly.

Line 317 "The simulated values of the depuration constant is 0.46 y$^{-1}$ whereas estimated from the field data for 2012-2015 in Fig.4b is 0.48 y$^{-1}$"

> 27. *Line 322: What do you mean with biomagification effect (see earlier comment) and how should that affect the CR in demersal fish mechanistically?*

**Answer:** The text was changed accordingly comments #8 and #27.

Line 24 "The estimated from model transfer coefficient from bulk sediment to demersal fish in the model for 2012-2020 (0.13) is larger than that to the deposit feeding invertebrates (0.07)."
Line 320 "Notice that the predicted transfer coefficient from bulk sediment to demersal fish for the period of 2012-2020 is approximately 0.13. This value is larger than that for deposit feeding invertebrates. The observed in this area BCF for demersal fish (flounders) in 2013-2015 is 0.9 m$^3$kg$^{-1}$, whereas the standard value of BCF for fish is 0.1 m$^3$kg$^{-1}$(IAEA, 2004) that confirms the importance of transfer of radiocaesium to demersal fish from the sediments."

> 28. *Line 323: Again inexact species and categorisation, there are several genus called rockfish, what is the scientific name? How does it fit into the classification? Coastal predator?*

**Answer:** We added in text scientific names for all organisms presented in Fig. 4.

Line 302 "The symbols in Fig. 4 are observation data for sea urchin (*Strongylocentrotus nudus*) (a), flounders (*Microstomus achne*, *Kareius bicoloratus*, *Pleuronectes yokohamae*) (b) and Japanese rockfish (*Sebastes cheni*) (c). The open and filled symbols in Fig. 4d are data for seabass (*Lateolabrax japonicas*) and fat greenling (*Hexagrammos otakii*), respectively."
Line 323 "Comparison of simulations with observations for a bottom predator (Japanese rockfish) in Fig. 4c shows a good agreement."

> 29. *Line 326: The legend to the figure could be put into figure text*

**Answer:** See answer on comment 28.

> 30. *Line 327-333: This is interesting results and probably support your approach, but it is messed up with inconsistent classifications. My suggestion that you first of all make a consistent classification, a clear description what that means and finally give examples of species in the area for each group. This should be done in methods, the you adhere*

*to the classification when you mention different species with common and scientific names (latin) . I know that it can never become clearcut where different species belong, but you tell at least the reader where they are in the model.*

**Answer:** See reworked text with description of each group of organisms in comment #2. The scientific species names are given in answers on comments#28 and #36.

*31. Line 334: ".. which are known with high uncertainity." Maybe better ... known to have a high…*

**Answer:** Done.

*32. Line 352: probably figure S3b not 3b*

**Answer:** The figure number was changed to Fig. S4b.

*33. Line 355: why this different numbers ?*

**Answer:** The text was changed accordingly:

"The maximum $^{137}$Cs concentration for zooplankton using the maximal value of $T_{0.5,i}$ was increased by a factor 2.7 compared with a case when the minimum value of $T_{0.5,i}$ was used. This factor for pelagic fish and coastal predator was in the range 2.4-1.7 whereas for the rest organisms it was smaller."

*34. Line 363: also probable wrong figure number*

**Answer:** The figure number was changed to Fig. S4d.

*35. Line 363-384: I would suggest to omit this part, there are assumptions and limits with different relevance in different parts of the world. From my horizon (responsible for dose assessments) I cannot see the point of this section. Omit it the also from conclusion*

**Answer:** Done.

*36. Line 385-410: If the modelling of the Baltic sea should be useful, this section should at least tabulate the drivers (fluxes over borders) and parameter values for the modelled box for the result. It is not reproducible with the current information. I am also missing general data on the bathymetry and which species are considered in the model foodweb.*

**Answer:** We added Table S2 with river runoff into the Baltic Sea and Table S1 with parameters for box 45 in the Baltic Sea.The scientific names for all organisms presented in Fig. 7 were also added.

Line 425 "The symbols in Fig. 7 are observation data for echinoderms (*Echinodermata*) (a), sprat (*Sprattus sprattus*) (b), European flounder (*Platichthys flesus*) (c) and Atlantic cod (*Gadus morhua*) (d)."

*37. Line 440: Polychaete feeding is that valid for the Baltic Sea*

**Answer:** We used information on diet of the European flounder (Platichthys flesus) from Gibson, R. N., Nash, R. D. M. Geffen, A. J., and Van der Veer, H. W. (eds.): Flatfishes: biology and exploitation. - Second edition Wiley-Blackwell, Chichester, UK, 2015. According Gibson and al. (2015) these fishes belong to group of "Polychaete and small crustacean feeders" (Table 11.1). This table provides more detailed information the European flounder in the Baltic Sea: it feeds by oligochaetes, amphipods, chironomids and smaller sizes harpacticoids. Text was changed accordingly.

Line 440 "Notice that European flounder diet in the Baltic Sea includes oligochaetes, amphipods, chironomids and smaller sizes harpacticoids. (Gibson et al., 2015).

*38. Line 459: As commented earlier the classification system needs to be reworked*
**Answer:** See answers on Comment #2.

*39. Line 464: Suggest to omit "strongly", it is not relevant for the Baltic Sea and I don't think I adds something more for Fukushima area.*
**Answer:** The text was changed accordingly.

Line 464: "The compartment model was applied to two regions (north western Pacific (NWP)) and the Baltic Sea) which were contaminated due to accidents on the Fukushima Dai-ichi and Chernobyl NPPs."

*40. Conclusion or discussion I am missing a more rigorous comparision between the Baltic Sea and Fukushima, otherwise I don't see the point include both in this paper.*
**Answer:** The text was added accordingly.

Line 387 "The Baltic Sea is an important case because of its transfer of $^{137}$Cs originating from the Chernobyl fall-out. It was chosen to verify the ability of the model with generic parameters to describe transfer processes in a semi-enclosed sea with very different oceanography."

Line 451 "The observed BCFs in this area for sprat, European flounder and Atlantic cod in 1990-2010 are 0.11, 0.14 and 0.15 m$^3$kg$^{-1}$, respectively. This is close to the standard value of BCF for fish 0.1 m$^3$kg$^{-1}$(IAEA, 2004) taking in account that waters in the Baltic Sea are brackish that affects the uptake rate of radiocaesium. These results essentially differ from the Fukushima case where BCF for demersal fish was an order greater confirming importance of transfer of radiocaesium from the sediments to demersal fish for that case."

Line 494-500 "The results of the application of POSEIDON-R with an extended dynamic model to the Baltic Sea which is semi-enclosed and filled by brackish waters are in good agreement with available measurements in the Baltic Sea. Unlike the highly dynamical off coast processes caused by eddy dominated currents in the Pacific Ocean where the FDNPP is located, weak water exchange with the North Sea and regular circulation in the Baltic Sea results in a slow quasi-equilibrium evolution of water-sediment-biota system. The Chernobyl case confirms that the standard parameterization of water-sediment exchange used in POSEIDON-R describes well the exchange processes for the Baltic Sea whereas in the Fukushima study the observed value of $^{137}$Cs decreases faster in the upper layer of the sediments than that the model predicts using the standard parameterization. In the Fukushima accident case the concentration of $^{137}$Cs in piscivorous fish decreases faster than in the coastal predators whereas in the Chernobyl case these concentrations decrease simultaneously. The obtained results demonstrate the importance of the benthic food chain in the long-term transfer of $^{137}$Cs from contaminated bottom sediments to marine organisms and the potential of a generic model for use in different regions of the World Ocean."

*41. No explanation in figure 1 what are the arrows, boxes, dotted lines, where are the explanations of the categories. What are the numbers? The dotted box with a waterbox outside? Water deposit what is that and why is that box outside ?*
**Answer:** The text and capture for Fig. 1 were changed accordingly. See answer on Comment #2.

*42. Figure 2. What are deep water boxes ? What are coastal box? Describe or give criteria or point to text where that is described*

**Answer:** See answer on Comment # 5.

*43. Figure 3 Explain what the legend means e.g correction of wha?t . kg of what drymatter ??*

**Answer:** See answer on Comment #21

*44. Figure 5 explain color-coding*

**Answer:** The text was added accordingly

"Figure 5. Compartment system of POSEIDON-R model for the North-Eastern part of the Atlantic Ocean, the North Sea and the Baltic Sea. The shaded boxes represent boxes divided on two vertical layers."

*45. Figure 6 is the concentration in bottom sediment for the bul sediment or organic fraction (the same question applies for Fukushima)*

**Answer:** The text was changed accordingly.

"Figure 3. Comparison between calculated and observed [137]Cs concentration in seawater (a) and in bulk bottom sediment (b) in the coastal box around the Fukushima Dai-ichi NPP."
"Figure 6. Comparison between calculated and observed [137]Cs concentrations in seawater (a) and in bulk bottom sediment (b) for box 45."

*46. FigS3 There no figtext for d)*

**Answer:** The text in the figure caption was corrected inserting (d).

**Response to Reviewer #2**

The authors are most grateful to the reviewer for his constructive criticism and suggestions. We have taken his remarks into account, and the paper has been revised accordingly.

*The paper is substantially corrected which makes reader more being easier to understand the result of the paper. Following points are recommended to reconsider or revise before publishing.*

*1) Citation of Matsumoto et al., 2015 should not be cited. The methodology in this paper (gross counts of Cs-137 energy band in whole body fish minus background counts without fish) contains bottom-up effect on net count by neglecting Compton effect from K-40 in fish flesh, which makes the turnover rate being overestimated.*

**Answer:** Text was changed accordingly. The reference on Matsumoto et al. (2015) was excluded from text.

Line 153 "The biological half-life data for fish flesh (Baptist and Price, 1962; Coughtrey and Thorne, 1983; Tateda, 1994,1997; Zhang et al., 2001) show variability in a large range (35-180 days) due to the differences between species and due to the differences in the experiment methodology."

*2) Rational for radiocesium turnover in bone etc. other than flesh should be shown by citation or theoretical assumption. It will help following further research by similar approach.*

According to Yankovich et al. (2010) the concentration of radiocaesium in muscle is 1.65 times higher than in the bone for marine fish. In combination with ratio between weight fractions of muscle and bone (0.845 and 0.135, respectively) the total amount of caesium in fish bone can be estimated only as 9% compared with muscle (90%) and organs (1%). Therefore, in a first approximation radiocaesium turnover in bones and organs was not considered. Text was added accordingly.

Line 187 "According to data from Yankovich et al. (2010) amounts of radiocaesium in flesh, bone and organs are 90%, 9% and 1%, respectively."

*Line 16 "released"*

**Answer**: Text was changed accordingly.

*Line 23 "evaluated as caused"*

**Answer**: Text was changed accordingly.

*Line 26 Is it sure to say? Better to refer only in main text as suggestion.*

**Answer**. "due to the biomagnification effect. " was excluded from text

*Line 33 "...suggest the substantial contribution..."*

**Answer**: Text changed accordingly.

*Table 3 Is there citation for the BHL of radiocesium in fish bone? Or theoretical assumption?*

**Answer: Text was added**

Line 161"The biological half-life for bone was estimated using data for $^{90}$Sr, which is mainly accumulated in bone. This value 
[revised manuscript text omitted]
). Notice that European flounder  diet in the Baltic Sea includes oligochaetes, amphipods, chironomids and smaller sizes harpacticoids (Gibson et al., 2015). The geometric mean  for the simulated-to-observed  ratios is 0.92 with a geometric standard deviation of 1.67 for a total number of observations $N$=70 in the whole Baltic Sea. Calculated  $^{137}$Cs concentration in the coastal predator (cod) also agree well with the measurements (Fig. 7d). The diet of Atlantic cod in shallow Western Baltic is diverse, including herring, sprat, *Gobiidae*, molluscs, various Polychaeta and crustaceans (Sparholt, 1994). Therefore for this basin the cod is considered as 'coastal predator' feeding by both pelagic and benthic preys. The geometric mean of the simulated-to-observed  ratios is 0.91 with a geometric standard deviation of 1.37 for a total number of observations $N$=95 in the whole Baltic Sea. The concentration of $^{137}$Cs in the coastal predators is greater than in piscivorous fish due to the additional benthic food chain included in the web (Fig. 7d).

[revised manuscript text omitted]

575  measurements. Unlike the highly dynamical off coast processes caused by eddy dominated currents in the Pacific Ocean where the FDNPP is located, weak water exchange with the North Sea  and regular circulation in the Baltic Sea results in a slow quasi-equilibrium evolution of  water-sediment-biota system. The Chernobyl case confirms that the standard parameterization of water-sediment exchange used in POSEIDON-R describes well

580  the exchange processes for the Baltic Sea whereas in the Fukushima study the observed value of $^{137}$Cs decreases faster in the upper layer of the sediments than that the model predicts using the standard parameterization. In the Fukushima accident case the concentration of $^{137}$Cs in piscivorous fish decreases faster than in the coastal predators whereas in the Chernobyl case these concentrations decrease simultaneously. In general, 
[revised manuscript text omitted]

[Figure]

**Figure 2.** The box system for the area close to Fukushima NPP. The shaded boxes represent the  deep-sea water boxes divided on three vertical layers. The NPPs are shown by filled circles. Coastal box around the FDNPP (marked by "F" is inside of box 90. Thick line limits the area of the Fukushima accident fallout.

[Figure]

**Figure 3.** Comparison between calculated and observed $^{137}$Cs concentration in seawater (a) and in bulk bottom sediment (b) in the coastal box around the Fukushima Dai-ichi NPP .The dashed line in (b) shows results of simulations using standard POSEIDON-R model, whereas solid line presents simulation with correction term in equation (S3)

[Figure]

**Figure 4.** Comparison between calculated and observed $^{137}$Cs concentration in deposit feeding invertebrate (a), demersal fish (b), bottom predator (c) and coastal predator (d) around the FDNPP.

[Figure]

**Figure 5.** Compartment system of POSEIDON-R model for  the North-Eastern part of the Atlantic Ocean, the North Sea and the Baltic Sea. The shaded boxes represent boxes divided on two vertical layers.

[Figure]

**Figure 6.** Comparison between calculated and observed $^{137}$Cs concentrations in seawater (a) and in bulk bottom sediment (b) for box 45.

[Figure]

**Figure 7.** Comparison between calculated and observed $^{137}$Cs concentrations in deposit-feeding invertebrate (a), non-piscivorous fish (b), demersal fish (c) and coastal predator (d) for box 45.

**Supplementary materials to the paper "Transfer of radiocaesium from contaminated bottom sediments to marine organisms through benthic food chain"**

**Poseidon-R model**

The mechanisms of radionuclide transfer in the POSEIDON-R model (Lepicard et al., 2004) are as follows. Activity entering the water column is transported by currents and turbulent diffusion and lost to bottom sediments through sorption on suspended particles which then settle out. The exchange of activity between the upper layer of the sediment and the water column is described as diffusion and bioturbation (modelled as a diffusion process). Activity in the upper sediment layer may diffuse downward but there is also an effective downward transfer via the continued sedimentation at the top of the sediment layers. Return of activity from the middle sediment to the top sediment occurs only through diffusion. The effective loss of activity from middle sediment to deep sediment arises from the continued deposition of sediment. A more detailed composition of the water column and its sediment layers, as well as its interaction with neighbouring volumes is shown in Fig. S1.

The POSEIDON-R equations are obtained by averaging the three dimensional transport equations for the dissolved radionuclide concentration $C_w$ (Bq·m$^{-3}$) and the concentration in the three layers of the bottom sediment. It is assumed that the activity in the water column is partitioned between the water phase and the suspended sediment material, resulting in the following relation:

$$C_{ss} = K_d C_w. \tag{S1}$$

where $C_{ss}$ (Bq·kg$^{-1}$) is the concentration of radioactivity sorbed by suspended sediment, $K_d$ is the radionuclide distribution coefficient (m$^3$·kg$^{-1}$). The equations read as follows.

For the water column layers:

$$\frac{\partial C_{w,i}}{\partial t} = \sum_j \left[ \frac{F_{ji}}{V_{w,i}} C_{w,j} - \frac{F_{ij}}{V_{w,j}} C_{w,i} \right] + \gamma_{0i} C_{w,(i,j,k-1)} - (\gamma_{1i} + \lambda) C_{w,i} + \frac{L_{t,i}}{h_i} \gamma_2 C_{s,1} + Q_{si}; \tag{S2}$$

for the upper sediment layer:

$$\frac{\partial C_{s,1}}{\partial t} = -(\gamma_2 + \gamma_3 + \lambda) C_{s,1} + \frac{h_i}{L_{t,i}} \gamma_{1,i} C_{w,i} + \frac{L_{m,i}}{L_{t,i}} \gamma_4 C_2, ; \tag{S3}$$

for the middle sediment layer:

$$\frac{\partial C_{s,2}}{\partial t} = -(\gamma_4 + \gamma_5 + \lambda) C_{s,2} + \frac{L_{t,i}}{L_{m,i}} \gamma_3 C_{s,1}. \tag{S4}$$

[Figure]

Fig.S1 Vertical structure and radionuclide transfer processes in the compartment of POSEIDON-R model. Arrows show exchange between boxes and layers.

Here subscript (0) denotes the water column, subscripts (1) and (2) denote the upper and middle sediment layer, respectively; $C_{w,i}$ is the box averaged concentration of radionuclide $C_w$ in the water column layer $i$; $C_{s,1}$ is the averaged concentration of radionuclide in the upper sediment layer; $C_{s,2}$ is the averaged concentration in the middle sediment layer; $\lambda$ ($y^{-1}$) is the radionuclide decay constant; $F_{ij}$ is the water flux ($t \cdot y^{-1}$) from box $i$ to box $j$; $V_{w,i}$ is the box volume ($m^3$); $h_i$ is the depth of the water box layer (m); $L_t$, $L_m$ are the depth (m) of top and middle bottom sediment layers respectively; $Q_{si}$ is the point source of the activity in box $i$ (Bq $y^{-1}$); $\gamma_0 \ldots \gamma_5$ are the coefficients, their values depend on the characteristics of the radionuclide and sediments.

For the surface water layer, the coefficients are as follows:

$$\gamma_{0i} = 0,$$

$$\gamma_{1i} = \frac{K_d SSW}{h_i(1 + K_d SS)},$$
(S5)

$$\gamma_2 = 0.$$

and for the layers in the water column below the surface water layer, the coefficients are defined as follows:

$$\gamma_{oi} = \frac{K_d SSW}{h_i(1 + K_d SS)},$$

$$\gamma_{1i} = \frac{K_d SS}{h_i(1 + K_d SS)},$$
(S6)

$$\gamma_2 = 0.$$

In the near bottom layer located at the bottom of the water column just above the bottom sediment, the coefficients are defined as:

$$\gamma_{oi} = \frac{K_d SSW}{h_i(1+K_d SS)},$$

$$\gamma_{1i} = \frac{K_d SSW}{h_i(1+K_d SS)} + \frac{1}{(1+K_d SS)}\frac{1}{L_b \min(L_b, L_t)} + \frac{K_d SSW}{(1+K_d SS)}\frac{B}{L_b \min(L_b, L_t)},$$

$$\gamma_2 = \frac{1}{R}\frac{D}{L_t \min(L_b, L_t)} + \frac{(R-1)}{R}\frac{B}{L_t \min(L_b, L_t)},$$

$$\gamma_3 = \frac{R-1}{R}\frac{SSW}{L_t(1-\varepsilon)\rho} + \frac{1}{R}\frac{D}{L_t \min(L_t, L_m)},$$

(S7)

$$\gamma_4 = \frac{1}{R}\frac{D}{L_m \min(L_t, L_m)},$$

$$\gamma_5 = \frac{(R-1)}{R}\frac{SSW}{L_m(1-\varepsilon)\rho},$$

where the coefficient $R$ is defined as:

$$R = 1 + \frac{\rho(1-\varepsilon)}{\varepsilon}K_d \qquad (S8)$$

Here $L_b$ (m) is the length scale of the bottom boundary layer, $SS$ is the different for each box concentration of suspended sediments (t·m$^{-3}$), obtained from observations or model simulation, $W_g$ is the settling velocity calculated as a function of suspended particles size; $SSW=SS·W_g$ is the fixed sediment flux (t·m$^{-2}$·yr$^{1}$); $D$ is the coefficient of vertical diffusion in the bottom; $B$ is the coefficient of bioturbation in the top bottom; $\varepsilon$ is the porosity of the bottom sediment; $\rho$ is the sediment density.

[Figure]

Fig. S2. The compartment system for the Northwestern Pacific. The shaded boxes represent the deep water boxes. The arrows with numbers show the compartments representing estuaries of large rivers (174 – the Yangtze River, 173 – the Huanghe River and 175 – the Han River. The NPPs are shown by filled circles. Letter "F" represent the Fukushima Dai-ichi NPP.

[Figure]

Fig. S3. Time variations of the annual deposition on the surface compiled from Nakano (2006) and Hirose et al (2008) (a) and the boundary values for the $^{137}$Cs concentration in the NW Pacific compiled from MARIS (2012) database and Kang et al. (1997) (b).

Table S1 The model parameters for coastal box around the FDNPP and box 45 in the Baltic Sea.

| Parameter | Coastal box (Fukushima case) | Box 45 (Baltic Sea case) |
|---|---|---|
| Volume, km$^3$ | 22.5 | 776.3 |
| Average depth, m | 50 | 31.4 |
| Water exchange rate with adjacent compartments, km$^3$yr$^{-1}$ | 150 | 4430 |
| Thickness of top sediment layer, m | 0.1 | 0.05 |
| Concentration of suspended sediments, kg/m$^3$ | $8 \cdot 10^{-2}$ | $1 \cdot 10^{-3}$ |
| Sedimentation rate, kg(m$^2$ yr)$^{-1}$ | $1 \cdot 10^{-2}$ | $7.5 \cdot 10^{-2}$ |
| Salinity, PSU | 35 | 15 |
| Sediment density, kg m$^{-3}$ | 2600 | 2600 |
| Vertical diffusion coefficient in bottom sediments, m$^2$yr$^{-1}$ | $3.15 \cdot 10^{-2}$ | $3.15 \cdot 10^{-2}$ |
| Bioturbation coefficient, m$^2$yr$^{-1}$ | $3.6 \cdot 10^{-5}$ | $3.6 \cdot 10^{-5}$ |
| Porosity of bottom sediments | 0.75 | 0.75 |
| $^{137}$Cs distribution coefficient, $K_d$, m$^3 \cdot$kg$^{-1}$ | 2 | 2 |

[Figure]

Fig. S4. Sensitivity indexes calculated for food uptake rate $K_1$ (a), biological half-life $T_{0.5}$ of $^{137}$Cs in the organism (b), water uptake rate $K_w$ (c) and for ratio of concentration of assimilated radioactivity from organic fraction of bottom sediment to the concentration of radioactivity of bulk bottom sediment $\phi_{org}$ (d).

Table S2 The river runoff into the Baltic Sea (Lepparanta and Myrberg, 2009.

| River box → Baltic Sea box | Rivers | Inflow (km$^3$·yr$^{-1}$) |
|---|---|---|
| 82 → 36 | Gota-alv + small rivers | 23 |
| 83 → 39 | All Danish rivers | 5 |
| 84 → 43 | Small German rivers | 9 |
| 85 → 45 | Oder + small rivers | 25 |
| 86 → 47 | Wisla + small rivers | 50 |
| 87 → 48 | Neman + small rivers | 30 |
| 88 → 59 | Motala strem + Swedish small rivers | 9 |
| 89 → 58 | Daugava + small rivers | 31 |
| 90 → 65 | Narva + small rivers | 20 |
| 91 → 66 | Kymijoki + small rivers | 13 |
| 92 → 67 | Neva | 79 |
| 93 → 71 | Dalalven + small rivers | 18 |
| 94 → 77 | Kokemenjoki + other Finnish small rivers | 25 |
| 95 → 78 | Angerman-alv + Indals-alv + smaller rivers | 47 |
| 96 → 80 | Ume-alv + smaller rivers | 22 |
| 97 → 81 | Kemijoki + Oulujoki + Lijoki + Torne-alv + Kalix-alv + Lule-alv + smaller rivers | 78 |

[Figure]

Fig. S5. Global atmosphere deposition rate of $^{137}$Cs on the Baltic Sea (HELCOM, 1995) (a) and release of $^{137}$Cs from Sellafield and La Hague reprocessing plants (HELCOM, 2009).

Table S3. Atmosphere deposition density of $^{137}$Cs in 1986 due to the Chernobyl accident (HELCOM, 1995)

| Basin | Deposition density, Bq m$^{-2}$ | Inventory, PBq | Boxes |
|---|---|---|---|
| North-Atlantic | 1000 | 35.4 | 3-34 |
| Kattegat | 1700 | 0.04 | 35-40 |
| Belt Sea | 1800 | 0.05 | 41-43 |
| Baltic Proper | 4500 | 0.82 | 44-57, 59-61 |
| Gulf of Riga | 5000 | 0.08 | 58 |
| Gulf of Finland | 15000 | 0.83 | 62-67 |
| Aland Sea | 72500 | 0.55 | 68-69, 71-72 |
| Archipelago Sea | 17300 | 0.04 | 70 |
| Bothnian Sea | 35000 | 1.94 | 73-79 |
| Bothnian Bay | 6900 | 0.31 | 80-81 |